# Astrocyte deletion of *Bmal1* alters daily locomotor activity and cognitive functions via GABA signalling

Olga Barca-Mayo[1,2], Meritxell Pons-Espinal[1], Philipp Follert[1], Andrea Armirotti[3], Luca Berdondini[2,*] & Davide De Pietri Tonelli[1,*]

Circadian rhythms are controlled by a network of clock neurons in the central pacemaker, the suprachiasmatic nucleus (SCN). Core clock genes, such as *Bmal1*, are expressed in SCN neurons and in other brain cells, such as astrocytes. However, the role of astrocytic clock genes in controlling rhythmic behaviour is unknown. Here we show that ablation of *Bmal1* in GLAST-positive astrocytes alters circadian locomotor behaviour and cognition in mice. Specifically, deletion of astrocytic *Bmal1* has an impact on the neuronal clock through GABA signalling. Importantly, pharmacological modulation of GABAA-receptor signalling completely rescues the behavioural phenotypes. Our results reveal a crucial role of astrocytic *Bmal1* for the coordination of neuronal clocks and propose a new cellular target, astrocytes, for neuropharmacology of transient or chronic perturbation of circadian rhythms, where alteration of astrocytic clock genes might contribute to the impairment of the neurobehavioural outputs such as cognition.

[1] Neurobiology of miRNA Lab, Neuroscience and Brain Technologies Department, Fondazione Istituto Italiano di Tecnologia, Via Morego 30, 16163 Genoa, Italy. [2] NetS3 Lab, Neuroscience and Brain Technologies Department, Fondazione Istituto Italiano di Tecnologia, Via Morego 30, 16163 Genoa, Italy. [3] D3 PharmaChemistry, Department of Drug Discovery and Development, Fondazione Istituto Italiano di Tecnologia, Via Morego 30, 16163 Genoa, Italy. * These authors contributed equally to this work. Correspondence and requests for materials should be addressed to O.B.-M. (email: olga.barca@iit.it) or to D.D.P.T. (email: davide.depietri@iit.it).

Animals have an internal timekeeping mechanism to anticipate daily changes associated with the transition of day to night[1], which is deeply involved in the regulation and maintenance of behavioural and physiological processes[2]. In mammals, the circadian system is organized in a hierarchy of multiple oscillators at organism, cellular and molecular level. At the organism level, the suprachiasmatic nucleus (SCN) is the central pacemaker at the top of the hierarchy, which integrates light information to ultimately regulate rhythms in gene expression, physiology and behaviour. At the cellular level, the SCN is composed of multiple oscillating neurons that are coupled to act as a single circadian unit[3], leading to coordinated circadian signalling outputs. At the molecular level, the circadian clock consists in the transcription- and translation-based interconnected feedback loops, in which the transcription factors BMAL1 and CLOCK drive the expression of *Per* and *Cry* genes, whose products lead to the inhibition of their own transcription[1]. Oscillations in abundance of those core clock proteins in the brain and peripheral tissues drive a cascade of transcriptional output genes that are not involved in the timekeeping mechanism itself, but underlie local behavioural and physiological process[4].

The SCN and other brain regions are composed of a heterogeneous population of cells, including astrocytes, which have a well-documented role in cooperating with presynaptic and postsynaptic neuronal elements to regulate communication events and behavioural processes. Although recent evidence suggests an involvement of astrocytes in the regulation of circadian rhythms in *Drosophila*[5,6] and in mammals[7,8], the role of clock genes in these cells has not been investigated.

It is currently unknown whether the regulation of astrocyte physiology by clock genes might contribute to the maintenance of neuronal rhythmic behaviour at cellular, tissue and organism level. Remarkably, SCN astrocytes express transporters for GABA (γ-aminobutyric acid), the principal neurotransmitter in the master pacemaker, and by up taking GABA from the extracellular space also synapses in the SCN function as 'tripartite' synapses[9]. Indeed, the circadian release of gliotransmitters, such as ATP[10] and the rhythmic expression of neurotransmitter transporters was also reported[11]. Identifying such a role for astrocyte clock genes will not only reveal a more complex cellular signalling in the brain than that considered so far but would also have significant implications for therapeutic research on transient perturbations in circadian rhythms (such as jet lag), adverse effects of shift workers and in disorders associated with circadian rhythms dysfunctions.

Here we report that astrocytic BMAL1 impacts the neuronal clock by altering GABAergic signalling. Importantly, this leads to altered circadian locomotor behaviour and to severe cognitive defects in mouse.

## Results

**Efficient deletion of *Bmal1* in SCN astrocytes.** To evaluate the role of the core clock gene *Bmal1* in astrocytes of adult mice *in vivo*, we generated a conditional Tamoxifen (TM)-inducible knockout mouse model (*Bmal1*[flx/flx], *Glast*-CreER[T2] +/− here referred to as *Bmal1*cKO), where Cre-recombinase is expressed under the control of the glutamate transporter *Glutamate Aspartate Transporter* (*Glast*) promoter, a widely accepted astrocyte-specific gene[12–14]. It has been previously shown that adult induction of CreER[T2] expressed from the *Glast* locus targets a large subset (60–80%) of astrocytes, corresponding in their frequency to those that endogenously express GLAST in cortex and striatum[12]. In contrast, Cre-mediated recombination in adult *Glast*:Cre-IRES-hrGFP mice has been shown to occur only in

0.1% of NEUN-positive neurons[12], thus confirming the astrocyte-specific recombination.

As the inducible form of Cre (CreER[T2]) cannot be localized by immunocytochemistry[12], to ascertain the Cre-mediated recombination we crossed *Glast*-CreER[t2] mice with a reporter mouse line, in which red fluorescent reporter Td-TOMATO is driven by CAG promoter, on Cre-mediated recombination of loxP sites. Two months after TM treatment, we analysed the *Glast*-Cre-mediated recombination and immunoreactivity for the astrocyte-specific markers glial fibrillary acidic protein (GFAP) or S100β in the SCN. *Glast*-Cre-Td-TOMATO reporter expression was intense ventrally and spread in central and dorsal SCN (Fig. 1a). We found that 49.53% and 46.51% of Td-TOMATO-positive cells co-localized with GFAP or with S100β, respectively, confirming that recombination occurred in astrocytes of the SCN (Fig. 1a and Supplementary Fig. 1a).

To evaluate the efficiency of *Bmal1* recombination in SCN astrocytes, we quantified the co-immunolocalization of BMAL1 and Td-TOMATO with GFAP or S100β in *Glast*-Cre-*Td-Tomato* (control) or *Bmal1*cKO-*Td-Tomato* animals at Zeitgerber (ZT) 0. Control animals expressed BMAL1 in 61.13% of Td-TOMATO-positive cells, whereas the number of Td-TOMATO cells expressing BMAL1 was reduced by ∼70% in *Bmal1*cKO-*Td-Tomato* mice (paired *t*-test, *P* = 0.0008; Fig. 1b,c). Similarly, the percentage of GFAP or S100β-positive astrocytes expressing BMAL1 was significantly reduced by 59.33% and 60.24%, respectively, in the SCN of *Bmal1*cKO-*Td-Tomato* mice (*P* = 0.0001 for GFAP and *P* = 0.0068 for S100β, paired *t*-test; Fig. 1d,e and Supplementary Fig. 1b).

Surprisingly, in mutant animals, the percentage of BMAL1-positive cells was also reduced by 51% in Td-TOMATO-negative cells (*P* = 0.02, paired *t*-test; Fig. 1c), suggesting that deletion of *Bmal1* in GLAST-positive astrocytes results in a global downregulation of BMAL1 in the SCN. Given the high expression of *Glast*-Cre-*Td-Tomato* by SCN astrocytes and the significant reduction of BMAL1 in the SCN of our mutants, we sought to evaluate circadian locomotor activity and cognition, both neurobehavioural outputs under circadian control, in *Bmal1*cKO mice.

**Altered circadian and cognitive phenotype in *Bmal1*cKO mice.** Two months after TM treatment (Fig. 2a), wheel-running activity of control (*Bmal1*[flx/flx])[15] and *Bmal1*cKO animals was used as an index of SCN circadian function[16,17]. Animals were entrained to a schedule of 12–12 h light–dark (LD) cycle for at least 8 days, before being transferred to constant darkness and then, re-entrained to a new 12–12 h LD cycle.

During the LD condition, the locomotor activity of *Bmal1*cKO mice was indistinguishable from that of control animals (Fig. 2b left panels and Fig. 2c), showing no differences in the periodicity or in the total average activity (Supplementary Fig. 2).

On release into constant darkness, control animals exhibited a daily rhythm with a period of ∼24 h (23.95 ± 0.08). Although this locomotor component was also observed in *Bmal1*cKO mice (24.04 ± 0.09), 71% of the mutants (5 of 7 animals) exhibited an additional activity component with periods of ∼12 h as shown by the Lomb–Scargle periodograms (Fig. 2b, right panels), suggesting a bimodal pattern of locomotor activity. There were no differences in the average activity or periodicity among genotypes (Fig. 2c and Supplementary Fig. 2). However, *Bmal1*cKO mice significantly delayed their active phase (11.70 ± 2.84 min per day) compared with control animals (6.88 ± 1.44 min; Fig. 2d, left panel).

After constant darkness, *Bmal1*cKO mice did not require a longer time to adjust their activity to a new LD cycle than control

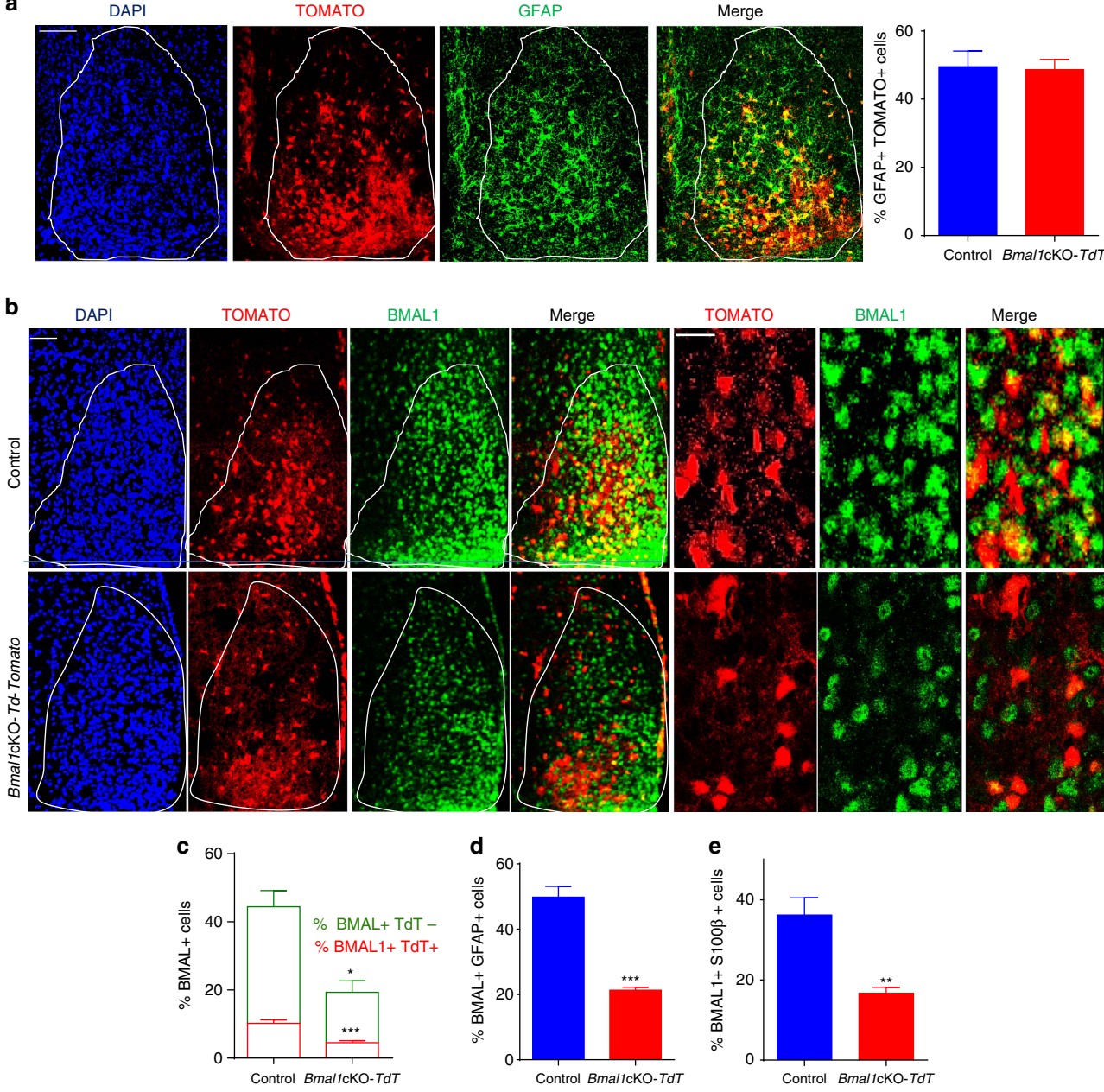

**Figure 1 | *Glast*-Cre-ER^t2 mediates astrocyte-specific deletion of *Bmal1* in the SCN.** (**a**) *Glast:*CreER^T2 mice drive the expression of reporter-Td-TOMATO in GFAP-positive SCN astrocytes. Representative micrographs of GFAP immunostaining in the SCN (dashed line) of control mice (4,6-diamidino-2-phenylindole (DAPI) in blue, TOMATO in red and GFAP in green). Scale bar, 50 μm. Quantification of the percentage of Td-TOMATO-positive cells that co-localized with GFAP in control (*Glast*-Cre-*Td-Tomato*) or *Bmal1*cKO-*Td-Tomato* animals is shown in the right panel. The value express mean ± s.e.m. (*n* = 4 animals per group). (**b**) *Glast:*CreER^T2-driven Td-TOMATO-positive SCN astrocytes express BMALl1. Representative micrographs of BMAL1 immunostaining in the SCN of control or *Bmal1*cKO-*Td-Tomato* animals in 12:12 h LD cycles (DAPI in blue, TOMATO in red and BMAL1 in green). Scale bars, 50 μm and 20 μm in the higher magnification images. (**c**) A 50% reduction of BMAL1 positive cells was observed in the SCN of *Bmal1*cKO mice compared with control animals (*Y* axis represents the percentage of total BMAL1-positive cells in the SCN). A 70% reduction of BMAL1-positive cells was observed in the population of Td-TOMATO-positive cells of *Bmal1*cKO compared with control animals (red, paired *t*-test, ***P < 0.001 versus control animals). A 51% reduction of BMAL1-positive cells in the population of Td-TOMATO-negative cells was found in *Bmal1*cKO compared with control animals (green, paired *t*-test *P < 0.05 versus control animals). The value express the means ± s.e.m. (*n* = 4 animals per group). (**d,e**) Percent of BMAL1-positive cells was significantly reduced in GFAP (**d**) or S100β (**e**) astrocytes in *Bmal1*cKO-*Td-Tomato* mice compared with control animals (paired *t*-test, ***P < 0.001 and **P < 0.01 versus control animals). The value express mean ± s.e.m. (*n* = 4 animals per group).

animals (Fig. 2b). However, *Bmal1*cKO mice showed a significant advanced onset (Fig. 2d, middle panel). Although mutant animals also showed an activity offset advance (Fig. 2d, right panel), the active period was significantly reduced in these mice

(13.05 h ± 0.34 for controls versus 11.35 ± 0.26 h for *Bmal1*cKO, paired *t*-test, *P* = 0.049).

None of the *Bmal1*cKO mice showed loss of rhythms, suggesting that *Bmal1* ablation in SCN astrocytes has a mild

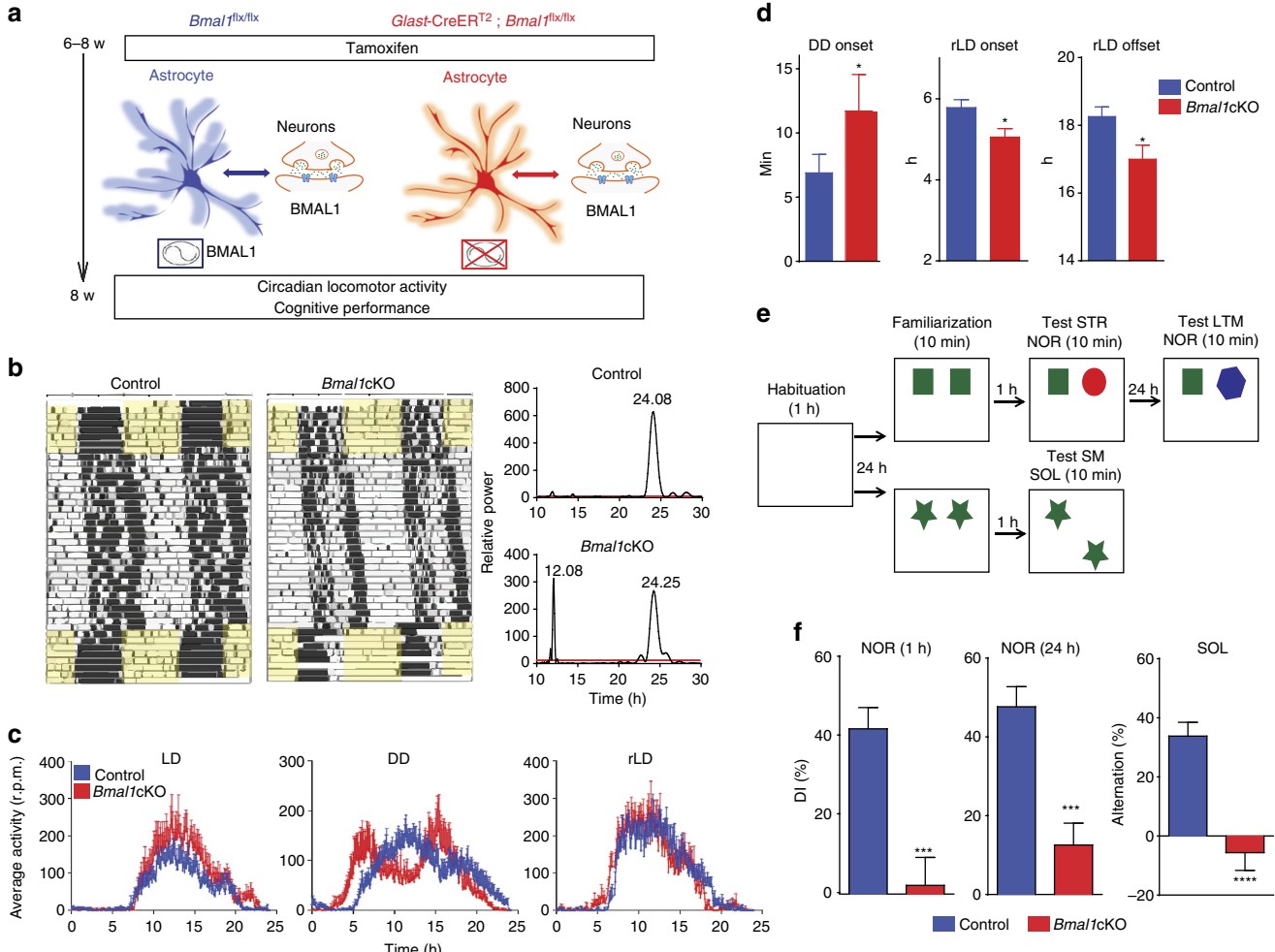

**Figure 2 | Circadian locomotor alteration and cognitive impairments on astrocyte-specific *Bmal1* deletion. (a)** *Bmal1*cKO and control mice were treated with TM and 6–8 weeks after treatment, circadian locomotor activity and cognitive tests were evaluated. **(b)** Representative actograms of control and *Bmal1*cKO mice during the 12:12 h LD, DD and the re-entrainment to a new LD cycle (rLD). Time of light is indicated by yellow shaded areas in the LD or rLD periods. The Lomb–Scargle periodograms (right panels) show the bimodal pattern of *Bmal1*cKO mice. **(c)** Activity waveforms under the LD, DD and rLD are shown for controls ($n = 8$) and *Bmal1*cKO ($n = 7$) mice. Activity counts are expressed as the average amount of activity in 5-minute bins. For LD and rLD, data are plotted with nighttime hours from 7 to 19 and given in Zeitgeber time (ZT), such that ZT0 (lights on) = hour 19. For DD, units on the abscissa are given in circadian time (CT) and mean activity was expressed as the average amount of activity in 5 min bins over each animal's circadian cycle. The value express the means + s.e.m. **(d)** Quantification of the activity onset in DD (left panel) indicated that *Bmal1*cKO mice significantly delayed their active phase compared with control animals (paired *t*-test, *$P < 0.05$ versus control animals). *Bmal1*cKO mice also showed an activity onset and offset advance in rLD cycles (middle and right panels) (paired *t*-test, *$P < 0.05$ versus control animals). The value express the means + s.e.m. **(e)** Diagram of the experimental design for the NOR and SOL tasks. **(f)** Performance on the NOR during 1 h retention and 24 h recall session and SOL in *Bmal1*cKO ($n = 9$) and control mice ($n = 10$). Paired *t*-test revealed a significant reduction in the DI between familiar and new object in *Bmal1*cKO in NOR tests and in the location of the object in the SOL test (paired *t*-test, ***$P < 0.001$ and ****$P < 0.0001$ versus control animals). The value express means + s.e.m.

impact on the clock. However, our results reveal that astrocytic BMAL1 is involved in the proper organization of daily locomotor activity in LD cycles. Moreover, the bimodal behaviour of *Bmal1*cKO mice, which might reflect the output of two circadian oscillators, suggests a potential role of astrocytes in the coupling of SCN oscillators that govern locomotor activities.

Perturbations of circadian rhythms in humans, such as in shift workers and resulting from jet lag[18], or in constitutive knockout mice for clock genes such as *Bmal1* − / − , have been associated with cognitive dysfunction[19]. We therefore evaluated a possible impact of astrocyte BMAL1 on cognition in our mutant mice. For this, the novel object recognition (NOR) test was used to assess short-term memory and long-term memory, by separating the sample and test phases by 1 and 24 h, respectively[20] (Fig. 2e). *Bmal1*cKO mice exhibited a significant reduction in the

discrimination index (DI) as compared with control mice after 1 and 24 h, thus indicating the impairment of both short- and long-term memory (Fig. 2f and Supplementary Fig. 3). In the spatial object location (SOL) task, only control animals but not *Bmal1*cKO, showed preferential exploration of the novel location, indicating compromised consolidation and object place recognition memory in mutant mice (Fig. 2f and Supplementary Fig. 3).

Altogether, our results indicate that the selective ablation of *Bmal1* in adult astrocytes is sufficient to alter daily locomotor activity and declarative memory in mice. These phenotypes might be dependent on astrocytic BMAL1 functions affecting gene expression in the SCN and/or rhythmic oscillations in cortical and hippocampal circuits involved on memory. Thus, we analysed whether rhythmic gene expression in the SCN, cortex and hippocampus were preserved in *Bmal1*cKO.

**Altered rhythmic gene expression in brain of _Bmal1_cKO mice.** Studies in _Drosophila_ showed that the involvement of astrocytes in the regulation of circadian rhythms[5,6] was mediated by a clock neuron peptide transmitter, pigment dispersing factor[6], which acts on a receptor similar to that for vasoactive intestinal polypeptide (VIP) in mammals. VIP is an oscillating neuropeptide with a light-dependent rhythm[21] that plays a well-defined role in coupling, synchronizing and phase-shifting rhythms within SCN neurons[22–24]. We investigated whether VIP expression was affected in the SCN of _Bmal1_cKO mice, by immunostaining. In particular, we compared the levels of VIP by immunofluorescence in the SCN of control and _Bmal1_cKO mice at ZT0 (when its expression peaks[21]) and ZT12 (Fig. 3a). Consistently with previous reports[21], VIP expression at ZT12 decreased compared with ZT0 in control mice (Fig. 3a). In contrast, VIP expression was not

downregulated in the SCN of _Bmal1_cKO mice at ZT12 (Fig. 3a). This result suggests repression of VIP by astrocytic BMAL1, leading to altered expression of this neuropeptide in mutant mice.

Next, we evaluated rhythmic oscillations of core clock genes in the cortex and hippocampus of _Bmal1_cKO mice. We found rhythmic expression of _Bmal1_, _Cry1_, _Per2_ and BMAL1 target _Dbp_[25] in control animals, with phases similar to those observed previously in pituitary gland and cortex[26–28]. However, these oscillations were impaired in _Bmal1_cKO mice (Fig. 3b and Supplementary Fig. 4a,b). As expected, BMAL1 levels were reduced by 30% in the cortex of _Bmal1_cKO mice and, notably, rhythmic expression of BMAL1 and PER2 dampened in the mutant mice (Fig. 3c and Supplementary Figs 4c and 5).

In our mice, BMAL1 depletion occurs in the majority of SCN astrocytes (as revealed by immunostaining of _Glast_-cre-driven

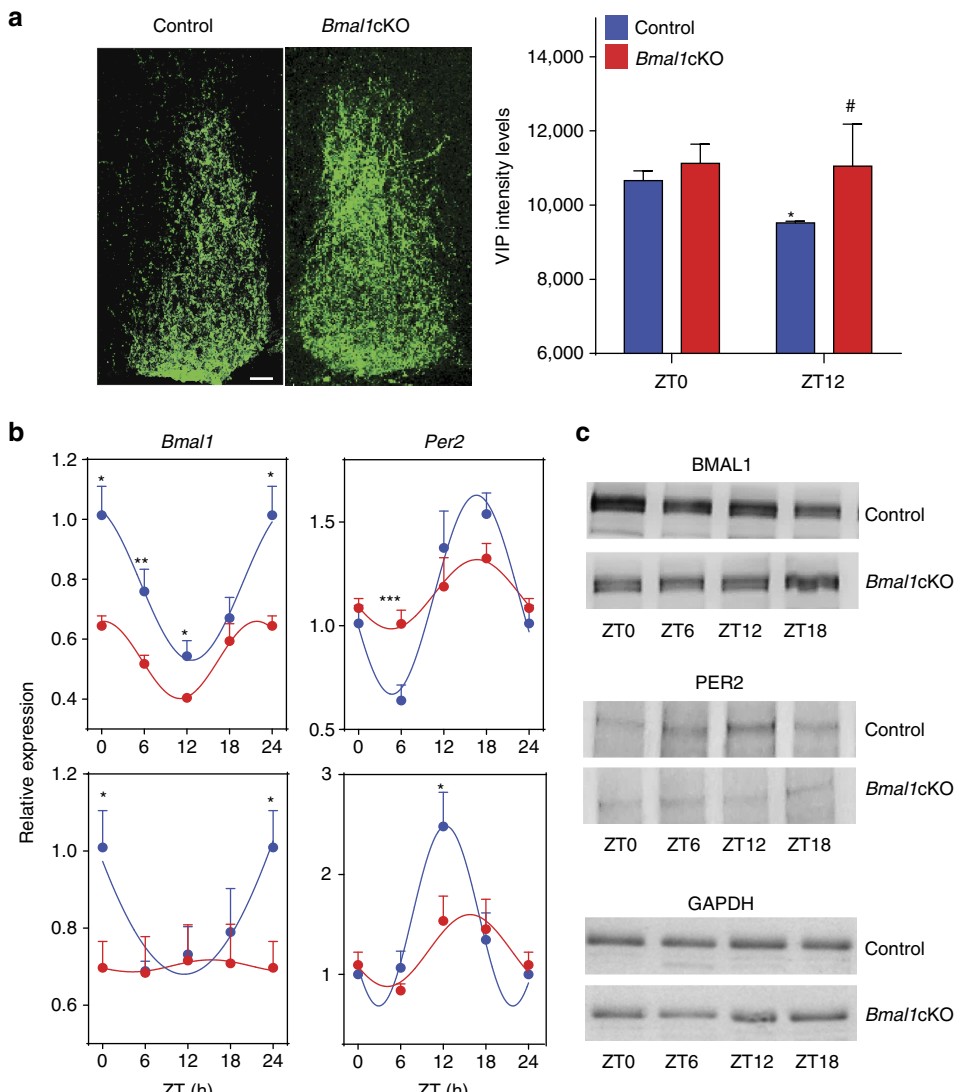

**Figure 3 | Altered VIP expression in the SCN and impaired cortical and hippocampal oscillations in _Bmal1_cKO mice.** (**a**) Representative micrographs of VIP immunostaining in the SCN of _Bmal1_cKO and control mice in 12:12 h LD cycles at ZT12. Quantification of fluorescence intensity demonstrates higher VIP levels in _Bmal1_cKO at ZT12 compared with control animals (paired t-test *P<0.05 versus ZT0 and #P<0.05 versus control animals). The value express the mean ± s.e.m. (n = 3 animals per group). Scale bar, 40 μm. (**b**) Analysis of _Bmal1_ and _Per2_ in the cortex (upper panels) and hippocampus (lower panels) of control (blue) and _Bmal1_cKO (red) mice, showing impaired rhythmic expression in mutant mice. Experimental data were cosine fitted. Samples were collected from mice under 12:12 h LD cycles. It is noteworthy that the ZT24 time point is the ZT0 time point, shown again. Means ± s.e.m. of five animals per group at each time point (paired t-test; *P<0.05, **P<0.01 and ***P<0.001 versus control animals). (**c**) Representative images of cortical BMAL1 and PER2 western blottings, showing no oscillation of those proteins in _Bmal1_cKO as compared ith control mice (n = 3 animals per group and time point).

Td-TOMATO with GFAP and S100β (Fig. 1 and Supplementary Fig. 1). However, we observed a significant reduction of BMAL1 in Td-TOMATO-negative SCN cells, a population likely to be embracing neuronal cells as well (Fig. 1c). This global reduction of BMAL1, together with the altered VIP expression in the SCN and the disruption of clock genes oscillation in both the cortex and hippocampus in Bmal1cKO mice, would be difficult to explain without considering the presence of intercellular communication between astrocytes and neurons. This scenario would be consistent with previous reports, indicating that astrocytes are competent circadian oscillators with temperature-compensated period[29] that can modulate clock gene expression of other cell types such as clock neurons or fibroblasts[30]. Thus, we hypothesized that BMAL1 function in astrocytes might be required to entrain circadian rhythmicity in neurons.

**Astrocytic BMAL1 is required for neuronal rhythmicity.** Impaired oscillations of core clock genes in the cortex and hippocampus of Bmal1cKO mice (Fig. 3 and Supplementary Figs 4 and 5) might depend on either a primary effect of astrocytic BMAL1 in the SCN, in local autonomous circuits or in both. As it is extremely difficult to discriminate between these possibilities, we investigated whether BMAL1 functions in astrocytes could affect the oscillations of core clock genes in cortical neurons. For that, we first set up a synchronization assay in astrocyte cultures by a short pulse (2 h) of Dexamethasone[31] (Fig. 4a). As expected, this treatment successfully induced rhythmic oscillation of core clock genes such as Cry1, Per2 and the BMAL1 target, Dbp, in control astrocytes (Astro Dexa, Fig. 4a,b). Next, we co-cultured synchronous astrocytes transfected with scramble small interfering RNA (siRNA) (Fig. 4a,b) or arrhythmic astrocytes (on Bmal1 knockdown, Fig. 4a,b) onto asynchronous primary cortical neurons, in physically separated layers (∼1.5 mm) but sharing the same culture media. We found that synchronous astrocytes systematically induced rhythmic expression of Bmal1, Cry1, Per2 and Dbp, as well as CRY1 in neurons (Fig. 4b,c and Supplementary Figs 6 and 7). Interestingly, we also found a significant advance in the acrophase of all the transcripts in synchronous astrocytes co-cultured with neurons, in comparison with synchronized astrocytes in isolated cultures (Fig. 4b and Supplementary Fig. 6a). In contrast, on Bmal1 knockdown, arrhythmic astrocytes failed to synchronize these transcripts and CRY1 in neurons (Fig. 4b,c and Supplementary Figs 6 and 7). Together, our results demonstrate that, by means of exchanged extracellular factor(s), astrocytic BMAL1 is required to entrain rhythmicity in neurons.

**GABAergic signalling mediates astrocyte–neuron communication.** We next sought to identify the extracellular factor that mediates astrocyte to neuron communication. It is widely accepted that astrocytes can dynamically regulate neuronal communication via the uptake of neurotransmitters (for example, GABA and glutamate) or by gliotransmitter release (such as D-serin and ATP)[32]. Among those molecules, glutamate is considered to be a major factor in transducing retinal photic information to the SCN via retinohypothalamic projection[33,34]. On the other hand, GABA, the principal neurotransmitter in the master pacemaker, is released in a daily rhythm within the SCN[35] and rhythmic levels were found in different brain regions[36]. Remarkably, SCN astrocytes express transporters for GABA and by uptaking GABA from the extracellular space also synapses in the SCN function as 'tripartite' synapses[9], where astrocytes can modulate neuronal activity and communication. Moreover, GABAergic signalling plays a role in tonic inhibition of neurons by modulating a continuous current dependent on extrasynaptic GABAA

receptors[37], whose are involved in memory[38]. Furthermore, astrocytic GABAergic signalling was shown to be involved in tonic inhibition in different brain areas[39]. Thereby, we hypothesized that glutamate and/or GABA might be the extracellular factor(s) mediating astrocyte to neuron communication.

To investigate this hypothesis, we tested whether glutamate or GABA can induce rhythmic oscillations of core clock genes in cortical neurons in vitro. We treated primary cortical neurons with either a 2 h pulse of glutamate (10 μM) or a pulse of GABA (100 μM) and we found that GABA, but not glutamate, was sufficient to entrain their rhythmic expression of Bmal1, Cry1, Per2 and Dbp in these cells (Fig. 5a).

Next, to determine whether GABA signalling can mediate astrocyte–neuron communication, we synchronized astrocytes with a short pulse (2 h) of Dexamethasone and co-cultured them with asynchronous cortical neurons (as in Fig. 4a), in the presence of the GABAA receptor blocker Bicuculline (30 μM). We found that the inhibition of GABAA receptor signalling prevents astrocyte-induced entrainment of clock gene oscillations in neurons (Fig. 5b). Remarkably, rhythmic expression of Per2, Cry1 and Dbp was maintained in astrocytes in the presence of Bicuculline (Supplementary Fig. 8), suggesting that GABAA receptor signalling is required to entrain rhythmic expression of clock genes in neurons, but not to sustain rhythmicity in astrocytes. Together, these results indicate that GABA, through GABAA receptor signalling, mediates astrocyte to neuron communication.

**Impaired GABA uptake on Bmal1 deletion in astrocytes.** Given that arrhythmic astrocytes fail to synchronize the neuronal clock (Fig. 4), we hypothesized that regulation of extracellular GABA might be impaired on Bmal1 deletion in astrocytes. To test this hypothesis, we analysed the expression of the GABA transporters (Gats), Gat1 and Gat3, which are localized to astrocytes, in the cortex and SCN of control and Bmal1cKO animals.

We found that neither Gat1 or Gat3 transcripts were oscillating in the cortex (Fig. 6a,b, left panels). However, Bmal1cKO mice showed a significant reduction of Gat1 (at ZT0 and ZT6) and Gat3 (at ZT0) as compared with control mice (Fig. 6a,b, left panels). We confirmed these results by immunostaining of GAT1 and GAT3 in cortex of control and mutant animals at ZT0 (Fig. 6a,b right panels and Supplementary Fig. 9a). As GAT1 is also expressed in axon terminals[40], to identify astrocyte-specific GAT1, this staining was performed in Bmal1cKO-Td-Tomato mice.

The reduced expression of astrocytic GAT1 and GAT3 in the cortex of Bmal1 cKO mice suggests a potential impairment in the clearance of extracellular GABA released by neurons. To verify this hypothesis we performed a GABA uptake assay in arrhythmic astrocytes on Bmal1 knockdown. Control or arrhythmic cortical astrocytes were treated with different doses of GABA and extracellular GABA levels were determined after 15 min by enzyme-linked immunosorbent assay (Fig. 6c). We found that GABA uptake was severely impaired in arrhythmic astrocytes (Fig. 6c), suggesting that astrocytic BMAL1 is required to avoid accumulation of extracellular GABA.

To further confirm this finding in vivo, we quantified GABA levels in the cerebrospinal fluid (CSF) of control and Bmal1cKO animals. Importantly, we found significantly higher GABA levels in the CSF of Bmal1cKO compared with control animals at ZT6 (Fig. 6d). We performed those measurements by pooling together CSF from several mice and following previously described liquid chromatography–tandem mass spectrometry (LC–MS/MS) protocols[41]. Although we could not find the reference values of

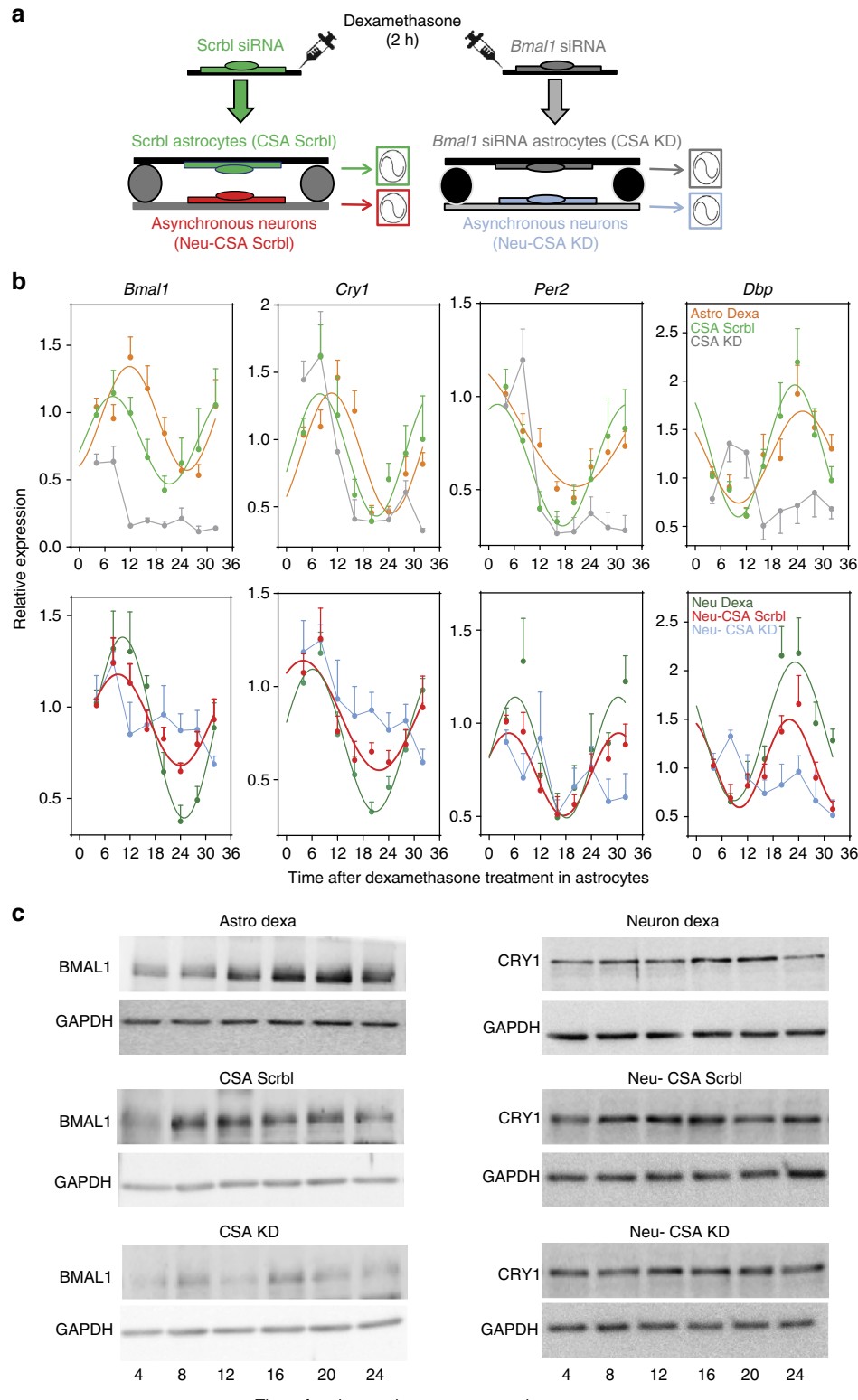

**Figure 4 | *Bmal1* knockdown in astrocytes suppresses entrainment of co-cultured cortical neurons *in vitro*. (a)** Primary cortical astrocytes were transfected with scramble (Scrbl) or *Bmal1* siRNAs. After 48 h, astrocytes were synchronized with 100 nM of Dexamethasone for 2 h (Astro Dexa). After washing, astrocytes were placed in co-culture with asynchronous cortical neurons without physical contact, but sharing the same culture media. (**b**) *Bmal1*, *Cry1*, *Per2* and BMAL1 target *Dbp* were analysed in astrocytes (upper panels) and neurons (lower panels) at the indicated time points by quantitative PCR. Graphs show the mean ± s.e.m. of the cosine-fitted curves from three experiments performed in triplicate. (**c**) Representative images of western blottings for BMAL1 in primary astrocytes (left panels) or CRY1 in co-cultured neurons (right panels), showing expression of BMAL1 in Dexamethasone-treated astrocytes in isolated cultures (top) or Dexamethasone-treated astrocytes in co-culture with asynchronous neurons (middle and bottom), on transfection with scramble siRNAs (CSA Scrbl) or on transfection with *Bmal1* siRNAs (CSA KD); (right panels) entrainment of CRY1 in cortical neurons after co-culture with Scrbl transfected synchronous astrocytes (Neu-CSA Scrbl) is not observed when co-culture is performed with arrhythmic astrocytes (Neu-CSA KD) (*n* = 2 independent experiments).

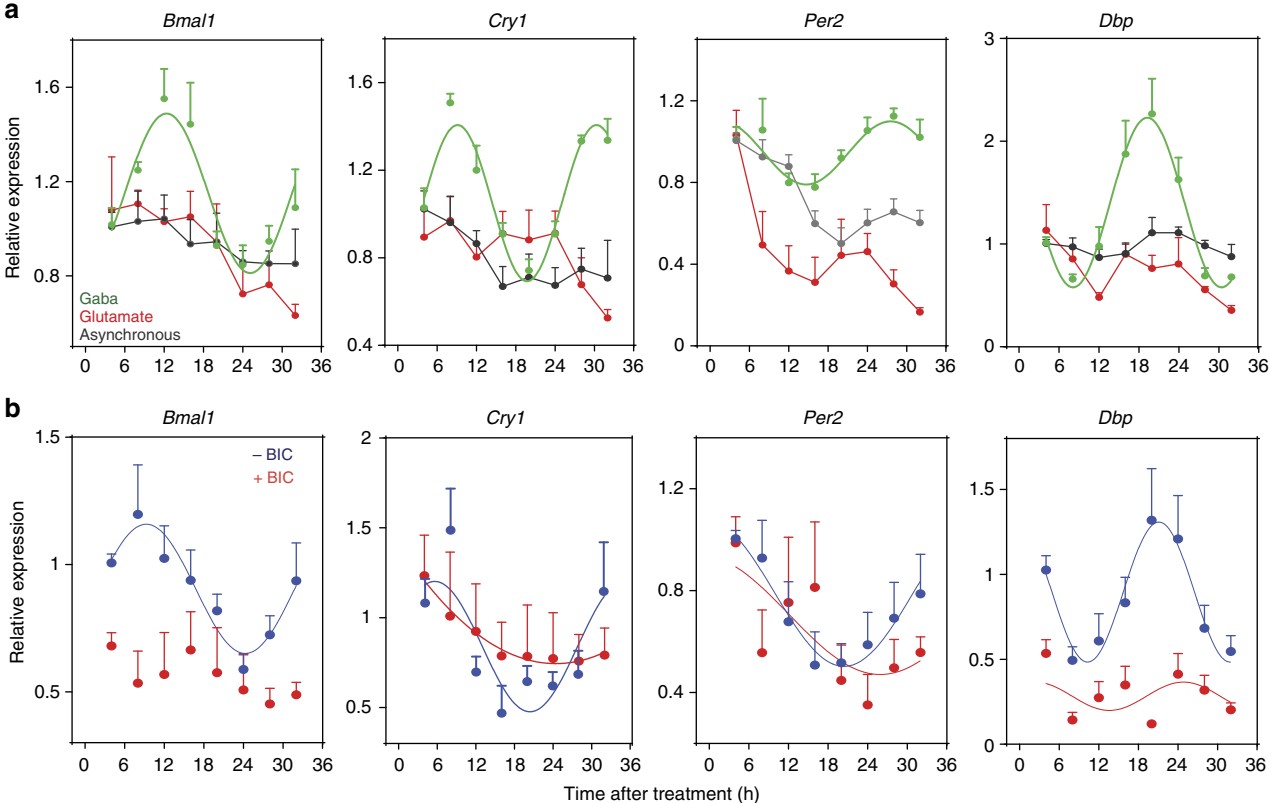

**Figure 5 | GABAA receptor signalling is sufficient and required to synchronize cortical neurons *in vitro*. (a)** Asynchronous cortical neurons were treated with a short pulse (2 h) of GABA (100 μM), glutamate (10 μM) or with vehicle as control. Cells were harvested at indicated time points and *Bmal1*, *Cry1*, *Per2* and *Dbp* were analysed by quantitative PCR (qPCR) and fitted to a cosinor curve. Each point reflects the means ± s.e.m. of three experiments performed in triplicate. **(b)** Primary cortical astrocytes were synchronized with Dexamethasone (100 nM) treatment for 2 h. After washing, astrocytes were placed in co-culture with asynchronous cortical neurons in the presence of Bicuculline (30 μM) or vehicle. *Bmal1*, *Cry1*, *Per2* and *Dbp* were analysed in neurons at indicated time points by qPCR and fitted to a cosinor curve. Graphs show the means ± s.e.m. of three independent experiments performed in triplicate.

the CSF GABA levels in mice, perhaps due to the intrinsic difficulty of the sampling in this animal model, our values are in line with those reported for humans (50–100 ng ml[−1]) and slightly lower than those reported for rats[42].

We also investigated whether astrocyte deletion of *Bmal1* might lead to an alteration of GATs in the central pacemaker. GAT1 is mostly expressed between the lobes of the SCN and around the third ventricle, whereas GAT3 is expressed evenly in the glial processes of SCN astrocytes[9]. Thus, we quantified the expression of GAT3 in the SCN in control and *Bmal1*cKO animals at ZT0. In control mice, GAT3 had a higher density in the dorsal and ventrolateral part of the SCN (Fig. 6e and Supplementary Fig. 9b). In contrast, this interregional distribution of GAT3 was lost in the SCN of *Bmal1*cKO mice and we found a significant reduction of GAT3 intensity in the dorsal part and an increase in the ventral SCN (Fig. 6e and Supplementary Fig. 9b).

All together, our results reveal that GABA can induce rhythmic oscillations of core clock genes in primary cortical neurons. Moreover, we also found that arrhythmic astrocytes, on *Bmal1* knockdown, cannot modulate their uptake of GABA. Indeed, *Bmal1*cKO mice showed reduced expression of GAT1 and GAT3, and elevated GABA levels in the CSF. Increased GABA in CSF of our mutants might not reflect local GABA levels in different areas of the brain. Indeed, the absolute concentrations of GABA in presynaptic cytosol, in vesicles and in the extrasynaptic space are unknown. However, the affinity constants of extrasynaptic GABA receptors may serve as a rough estimate of background

concentrations (0.2–2.5 μM)[43], and in rat, direct measurements from CSF yielded similar or slightly higher values[42]. Thus, the CSF levels of GABA might reflect the concentrations of this neurotransmitter at the extrasynaptic space.

Our results suggest that BMAL1 in astrocytes might play a fundamental role in maintaining extracellular GABA levels and/or rhythms in a range compatible with neuron synchronization. We postulate that this scenario could be involved in the cognitive impairments that we observed in *Bmal1*cKO mice. In fact, this would also be in agreement with evidence showing that the arrhythmic SCN of the Siberian hamster leads to an over-inhibition of synaptic circuits involved in memory. Indeed, the cognitive deficits of those animals were completely restored after administration of GABAA receptor antagonists or on SCN removal[44–46].

On the other hand, GABA has been shown to transmit phase information between the ventral and dorsal oscillators of the SCN[47]. This evidence, together with the loss of the interregional distribution of GAT3 expression in the SCN of *Bmal1*cKO mice, suggests that altered GABA-mediated coupling among the SCN oscillators might underlie the bimodal pattern of locomotor activity of our mutants.

**GABAergic antagonists rescue behaviour of *Bmal1*cKO mice.** To verify our hypothesis that altered GABA signalling might lead to the circadian locomotor and declarative memory phenotypes of *Bmal1*cKO mice, we administered GABAA receptor

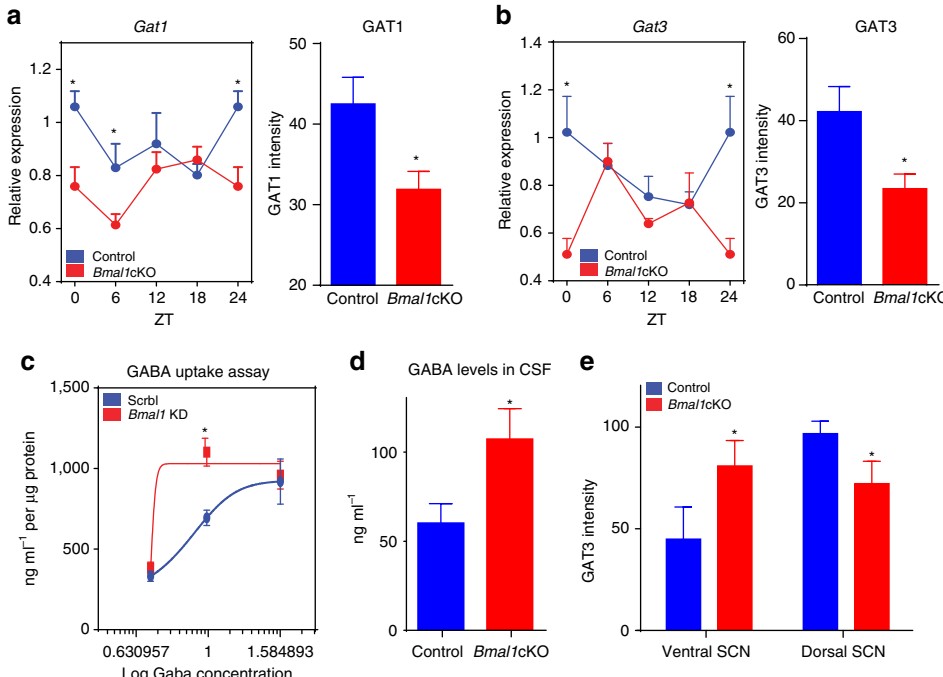

**Figure 6 | Alteration of GATs and GABA uptake on deletion of astrocytic Bmal1.** (**a**,**b**) Left panels: Gat1 (**a**) and Gat3 (**b**) in the cortex of control (blue) and Bmal1cKO (red) mice, showing reduced expression in Bmal1cKO. Samples were collected from mice under 12:12 h LD cycles. It is noteworthy that the ZT24 time point is the ZT0 time point, shown again. Means ± s.e.m. of five animals per group at each time point (paired t-test *$P < 0.05$ versus control animals). (**a**,**b**) Right panels: quantification of fluorescence intensity of GAT1 (**a**) and GAT3 (**b**) demonstrates decreased expression of GAT1 and GAT3 in Bmal1cKO-Td-Tomato or Bmal1cKO, respectively, at ZT0 compared with control animals. Means ± s.e.m. of four animals per group (paired t-test; *$P < 0.05$ versus control animals). (**c**) Primary cortical astrocytes were transfected with scramble (Scrbl) or Bmal1 siRNAs (Bmal1 KD) and after 48 h, were treated with 5, 10 or 40 µM of GABA for 15 min. GABA concentration in the extracellular medium was measured by enzyme-linked immunosorbent assay (ELISA). Graph shows the means ± s.e.m. of two independent experiments performed in triplicate (paired t-test; *$P < 0.05$ versus control astrocytes).
(**d**) Determination of GABA levels CSF of Bmal1cKO and control animals at ZT0 by ultra performance LC–MS/MS. Bmal1cKO mice showed significantly higher levels of GABA in the CSF than control animals. A total number of 14 controls and 10 Bmal1cKO animals were used for this experiment. Two or three individual animals were pooled into final samples ($n = 5$). Graph shows the means ± s.e.m. (paired t-test; *$P < 0.05$ versus control animals).
(**e**) Quantification of fluorescence intensity of GAT3 in the SCN, demonstrates lower levels in the dorsal part and increased levels in the ventral SCN in Bmal1cKO at ZT0 compared with control animals. Graph shows the means ± s.e.m. ($n = 4$ per group, paired t-test; *$P < 0.05$ versus control animals).

antagonists Pentylenetetrazole (PTZ) or Picrotoxin (PTX) (0.3 mg kg$^{-1}$ per day) to control and Bmal1cKO mice at ZT6, for 10 days (Fig. 7a), following previously reported protocols[45,46,48]. We then analysed the circadian locomotor activity and performed again the cognitive tests, as described above (Fig. 2).

Consistent with previous reports[49], PTZ treatment did not alter neither the average activity nor the periodicity of control mice in any of the lighting conditions (LD, dark–dark (DD) and re-entrainment to a new LD cycle; Fig. 7b,c and Supplementary Fig. 10). Similarly, we observed no significant differences between PTZ-treated and untreated Bmal1cKO mice. Indeed, the periodicity and the average activity was not different among control and Bmal1cKO animals on treatment with PTZ (Supplementary Fig. 10).

Remarkably, PTZ treatment rescued the bimodal pattern of locomotor activity of Bmal1cKO mice in DD as shown by the Lomb–Scargle periodograms (Fig. 7b, lower panels). This finding suggests that astrocytic BMAL1 couples the oscillators in the SCN through GABAA receptors. In this context, we hypothesize that the loss of the interregional distribution of GAT3 expression in the SCN of Bmal1cKO mice might alter local GABA levels to uncouple the oscillators in the central pacemaker. Moreover, we found that on PTZ treatment, the activity onset of Bmal1cKO in DD was not different from PTZ-treated control animals (Fig. 7d, left panel), thus restoring the delayed onset of activity observed in Bmal1cKO (11.70 ± 2.84 min per day for Bmal1cKO mice versus

5.38 ± 0.83 min for PTZ-treated Bmal1cKO). Similarly, the advanced activity onset and offset of Bmal1cKO mice in rLD were not observed on PTZ treatment and were not different from the PTZ-treated control animals (Fig. 7d, middle and right panels).

Declarative memory was also analysed in control and Bmal1cKO mice on PTX or PTZ treatment by subjecting animals to the NOR and novel object location tasks. Interestingly, we found that these treatments also re-established cognitive functions in Bmal1cKO mice to normal levels, whereas, similar to previous reports, did not augment the DI in control animals[45,48] (Fig. 7e and Supplementary Fig. 11). This result implies that mnemonic deficits observed in Bmal1cKO mice arise from specific abnormalities in declarative memory that are rescued by drug effects within the circuits involved, rather than to some nonspecific effects of the drugs.

We postulate that loss of BMAL1 function in astrocytes results in altered GABA levels, leading to the over-inhibition of the circuits involved in learning and memory and to the uncoupling the SCN oscillators. Consistently, administration of GABAA receptor antagonists restores the circadian locomotor activity and the cognitive functions of Bmal1cKO mice.

## Discussion
This study is the first demonstration of a role for astrocytic BMAL1 in the modulation of circadian locomotor behaviour and

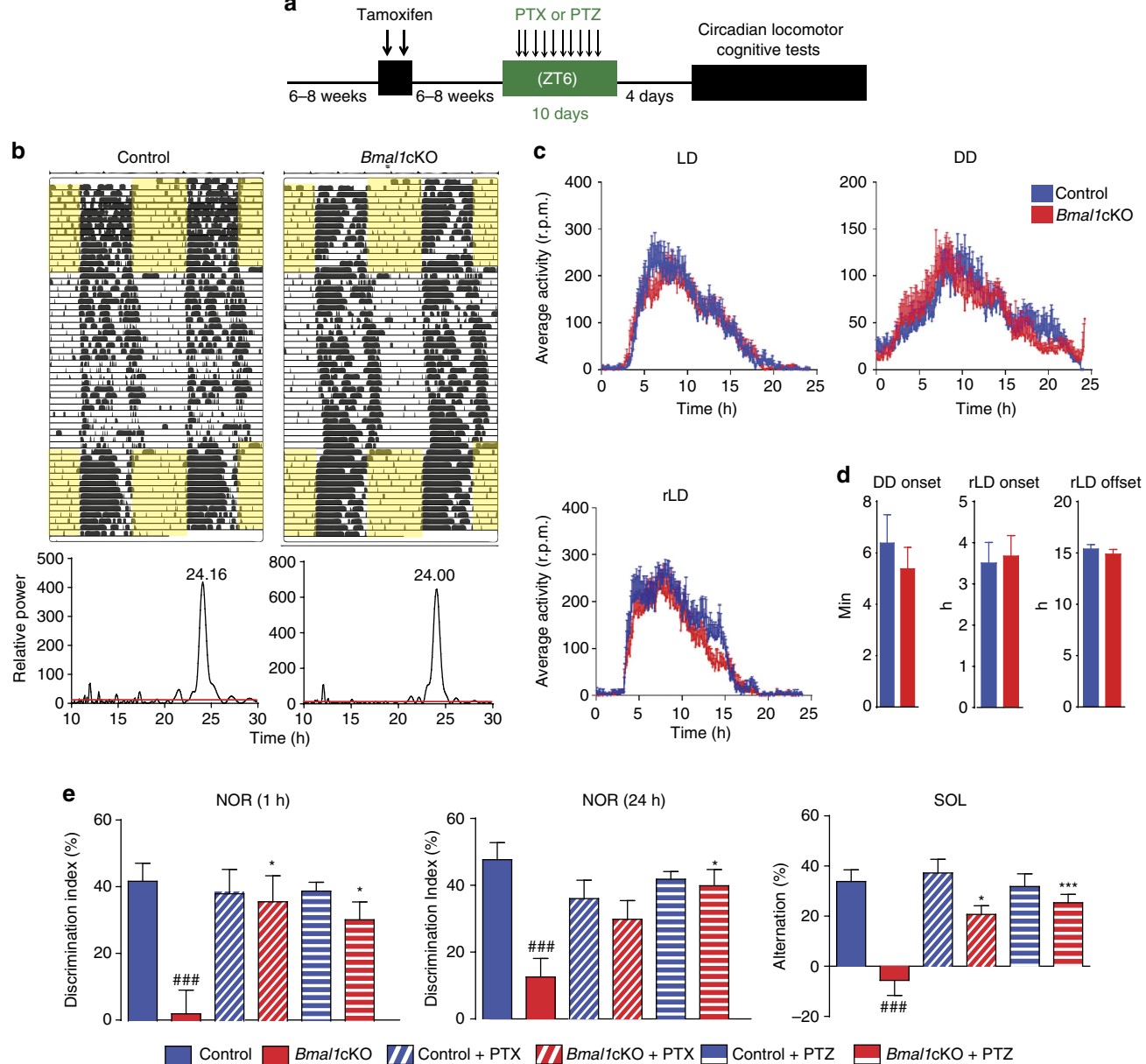

**Figure 7 | GABAA receptor antagonists rescue the behavioural phenotypes of *Bmal1*cKO mice.** (**a**) *Bmal1*cKO and control mice received, 6–8 weeks after TM treatment, one daily injection of (**b–d**) PTZ or (**e**) PTX at ZT6 for 10 days, before being subjected to (**b–d**) wheel-running activity analysis or (**e**) cognitive tests. (**b**) Upper panel: representative actograms of control and *Bmal1*cKO mice during the 12:12 h LD, DD and rLD cycles. Time of light is indicated by yellow shaded areas in the LD or rLD periods. (**b**) Lower panels: the Lomb–Scargle periodograms show the rescue of the bimodal pattern of *Bmal1*cKO mice by PTZ treatment. (**c**) Activity waveforms under the LD, DD and rLD are shown for controls (*n* = 9) and *Bmal1*cKO (*n* = 10) mice. Activity counts are expressed as indicated in Fig. 2c. The value express the means + s.e.m. (**d**) Quantification of the onset of activity in DD (left panel), in rLD (middle panel) and offset in rLD (right panel), indicating no differences among PTZ-treated *Bmal1*cKO and PTZ-treated control animals (paired *t*-test). The value express the means + s.e.m. (**e**) Performance on the NOR test during 1 h retention and 24 h recall session, and SOL task in Bmal1cKO (*n* = 10) and control mice (*n* = 9). No significant differences were found in the DI between familiar and new object in PTZ- or PTX-treated Bmal1cKO mice in NOR tests and in the location of the object in the SOL test, compared with control animals. The value express the means + s.e.m. (two-way analysis of variance, ###*P* < 0.001 versus control animals: *P* < 0.05 and ***P* < 0.001 versus untreated animals (that is, with neither PTZ or PTX).

cognitive functions. Specifically, here we show that deletion of Bmal1 in astrocytes has an impact on the neuronal clock through GABA signalling. Importantly, pharmacological modulation of GABAA-receptor signalling completely rescued the behavioural phenotypes of *Bmal1*cKO mice.

Remarkably, our study promotes *Bmal1*cKO mice as a valuable *in vivo* tool to model human pathologies related to alterations in the circadian system. The well-documented adverse effects of human dysrhythmia on cognition and declarative memory[18]

stands in stark contrast with the marginal effect of clock gene knockouts and SCN lesions in rodents. One critical factor that often gets overlooked in translating animal studies to human conditions is the fact that human dysrhythmia occurs, while the SCN circuitry remains intact, both genetically and structurally. Thus, our model is well suited for functional studies of the circadian system, because it allows acute adult disruption of astrocytic BMAL1, while avoiding functional abnormalities or compensations that might occur during development.

Our results indicate that deletion of *Bmal1* in a sub-population of astrocytes is sufficient to affect globally the neuronal clock in the brain. The anatomical properties of astrocytes are crucial to understand this finding. Indeed, cortical and hippocampal astrocytes are organized in structurally non-overlapping domains *in vivo*[50–52], where one astrocyte interacts with, an average of, four cortical neuron cell bodies and between 300 and 600 dendrites[50]. Moreover, astrocytes form a syncytium via gap junctions that is required for the circadian patterns of neuronal activity in the SCN[8]. Thus, although gap junctions mediate intercellular coupling among astrocytes, our data indicate that astrocytes regulate neuronal clock by a mechanism involving GABA and GABAA receptor signalling.

Although our results show a global reduction of BMAL1 in the SCN, some of the cells that retain BMAL1 expression might underlie the fact that none of the *Bmal1*cKO mice analysed showed arrhythmic locomotor behaviour. This finding is therefore consistent with different studies showing that partial deletion of *Bmal1* (refs 53,54) or partial lesions of the SCN[17,55–59] did not result in loss of rhythms. Whether deletion of *Bmal1* in all SCN astrocytes will drive to complete loss of daily locomotor activity is an unresolved question that is difficult to address. Indeed, astrocytes are widely distributed throughout the nervous system and express brain-region-specific genes. Thus, the selection of regulatory elements to target all astrocytes *in vivo* is almost impossible with today's tools.

However, based on our results, a potential contribution of intrinsic astrocyte clock to the global timekeeping system to finely coordinate autonomic circadian rhythms and associated neurobehavioural outputs cannot be discarded. In fact, *in vitro* studies indicate that astrocytes are competent circadian oscillators with a temperature-compensated period[29], which can modulate clock gene expression of other cells types such as clock neurons or fibroblasts[30]. Indeed, the circadian release of gliotransmitters, such as ATP[10], and the rhythmic expression of neurotransmitter transporters[11] indicate that astrocyte physiology is regulated by circadian rhythms. Thus, together with those findings, our study suggests that astrocyte rhythmic modulation of neurotransmission might be required for the correct organization of the hierarchy of oscillators at the cellular, tissue and organism level. Consistently, the behavioural phenotype of *Bmal1*cKO mice was completely rescued, on modulation of GABAA-receptor signalling, demonstrating a physiological regulation of neuronal function rather than a neural degeneration-induced phenotype.

*Bmal1*cKO mice showed impairments in restricting the locomotor activity to day or night when re-entrained to the new LD cycle. As *Glast* is expressed in the retinal Müller glia, it might be argued that this could be due to a secondary effect of altered retinal sensory inputs. However, this is unlikely as the circadian locomotor activity of *Bmal1*cKO and control mice is indistinguishable in LD cycles. On the other hand, it is widely accepted that glutamate signalling plays a crucial role in the transduction of retinal photic information to the SCN[33,34]. As in our mouse model *Bmal1* is conditionally deleted by *Glast*-driven Cre in astrocytes, a potential alteration in glutamate signalling might underlie the impaired light-mediated resetting of the circadian clock of *Bmal1*cKO mice. However, the phenotype of our mutants was restored on GABAA antagonist treatment, suggesting that altered glutamate signalling might not be the prime cause of this phenotype of *Bmal1*cKO.

Remarkably, astrocytic BMAL1 might have crucial implications for the synchronization of the internal clock. We found a bimodal pattern of activity when *Bmal1*cKO mice were released in constant darkness, suggesting uncoupled circadian oscillators governing locomotor activities. The loss of interregional distribution of GAT3 in the ventral and dorsal SCN of our mutants suggest that altered GABAaergic signalling might underlie this phenotype. Consistently, GABA has been shown to transmit phase information between the ventral and dorsal oscillators of the SCN[47]. Indeed, we found that GABAA antagonist treatment restores the bimodal locomotor behaviour of *Bmal1*cKO mice. However, we cannot discard an involvement of VIP on this phenotype, as it was found that administration of this neuropeptide increase the frequency of GABA inhibitory postsynaptic currents in clock neurons[35]. Consistently, hyperpolarized resting membrane potential of SCN cells was found in the $vpac_2^{-/-}$ mice[60]. Moreover, it was recently reported that VIP neurons evokes functional GABAergic responses, through GABAA receptor in the SCN[61]. Thus, we postulate that the contribution of astrocyte BMAL1 to the stable phase of the SCN clock is very likely to be a balance between VIP and GABAA receptor-mediated synchronization.

It was reported that the differential day/night glial coverage of VIP neurons dendrites is important to facilitate entrainment to light and to the pacemaker-resetting mechanism[62,63]. The finding that *Bmal1* deletion in GLAST-positive astrocytes alters VIP expression in the SCN is consistent with the intense expression of Td-TOMATO reporter in the ventral SCN, where VIP neurons are located. Similarly, studies in *Drosophila* showed that conditional, glial-specific genetic manipulations affecting membrane (vesicle) trafficking or calcium signalling lead to circadian arrhythmicity[5,6]. This behavioural phenotype was correlated with effects on a clock neuron peptide transmitter such as pigment dispersing factor[6], acting on a receptor similar to that for VIP in mammals. As VIP rhythm is driven by the LD cycle and not by the circadian clock, the mechanism by which astrocytes deficient in *Bmal1* alters VIP levels in our case remains unknown.

The phenotype of our mice might be dependent on static versus oscillating BMAL1 function in astrocytes. Ultimately, the function of clock proteins is never completely disengaged from their oscillation, as the BMAL1/CLOCK DNA binding shows clear circadian variation[64]. However, a major challenge, which holds true for most of the studies involving genetic manipulations of core clock genes, involves distinguishing the specific importance of circadian oscillation versus the 'static' function of clock genes. Despite this fact, our results together with the elegant studies in *Drosophila*[5,6] not only emphasizes the conservation of cellular and molecular mechanisms that regulate behaviour in mammals and insects but also demonstrate the capacity for astrocyte-to-neuron signalling in the circadian circuitry.

In conclusion, our study reveals a crucial role of astrocytic BMAL1 for the coordination of neuronal clocks and opens new avenues to understand the physiology of the timekeeping system and to treat or prevent disorders associated with them, based on astrocyte targeted drugs.

## Methods

**Animals and treatments.** Mice were housed with *ad libitum* access to food and water, and kept on a 12 h (07:00–19:00 h) LD cycle, in a room maintained at 21 °C at the animal facility of the Istituto Italiano di Tecnologia, Genoa, Italy. All experiments and procedures were approved by the Italian Ministry of Health (Permit Number 214/2015-PR) and local Animal Use Committee, and were conducted in accordance with the Guide for the Care and Use of Laboratory Animals of the European Community Council Directives and of Italian Ministry of Health.

*Bmal1*flx/flx mice on a pure C57/Bl6 background (Stock 007668, B6.129S4(Cg)-Arntltm1Weit/J, Jackson Laboratories) were bred with heterozygous *Glast*-CreER[T2] on a mixed C57/Bl6/Sjl/129 background (from M. Götz (Physiological Genomics, Biomedical Center, Ludwig-Maximilians-University Munich, Germany)), to obtain heterozygous *Bmal1*flx/flx *Glast*-CreER[T2] mice. Those heterozygous mice were then crossed with *Bmal1*flx/flx mice to obtain *Glast*-CreER[T2]; *Bmal1*flx/flx (*Bmal1*cKO). Bmal1cKO and control animals (*Bmal1*flxflx) were obtained from crosses between *Glast*-CreER[T2]; *Bmal1*flx/flx and *Glast*-CreER[T2] animals.

Six to 8 weeks old *Glast*-CreERT2; *Bmal1*[flx/flx] (*Bmal1*cKO) and controls (*Bmal1*[flxflx]) were treated with TM dissolved in corn oil. Animals received 5 mg per day for 2 days by oral gavage[12]. All behavioural studies and tissue samples were performed or collected, respectively, after 6–8 weeks of TM treatment from both male and female mice, except for the circadian locomotor behaviour where only male mice were analysed.

PTZ and PTX were dissolved in saline (NaCl) and administered to *Bmal1*cKO ($n = 10$) and control mice ($n = 9$) 0.3 mg kg$^{-1}$ per day by intraperitoneal injection (injection volume was 0.1 ml per 30 g body weight) at the middle of the light phase (ZT6) $\pm 1$ h. The chronic treatment protocol involved daily doses of PTZ or PTX for 10 days, followed by 2–4 days without treatment before assessment of learning and memory (NOR and novel object location) tests, or to wheel-activity monitoring. This protocol and dosage of PTX or PTZ normalize memory performance of Down syndrome mice models, from 1 week to 2 months (m) after the treatment[48].

Mice were randomized into the experimental groups based on their body weight.

**Mouse and behavioural activity monitoring.** Four to 5 months old male *Bmal1*cKO ($n = 7$) and control mice ($n = 8$) were single-housed in cages equipped with running wheels (ENV-044; Med Associates, Inc.). Mice were adapted to the wheel for 3 days in standard LD cycles (12:12 h, lights on at 07:00 h) and the experiment started under these conditions during at least 7 days. Then, mice were released in constant darkness for 3–4 weeks in isolated black cages (developed in collaboration with Tecniplast Spa, Italy) followed by, at least, 8 days under LD cycles (12:12 h, lights on at 07:00 h). Running wheel activity was recorded in 5 min bins by Wheel Manager software (SOF-860; Med Associates, Inc.). The data obtained were analysed with Actogram J.

**Cognitive tests.** Control and *Bmal1*cKO mice were housed two to five per cage and were given *ad libitum* access to food and water. They were kept on a 12 h (07:00–19:00 h) LD cycle in a room maintained at 21 °C. All tests were conducted during the light cycle (ZT 3–7). Mice were habituated to handling and transport from the colony room to the behavioural room for 3 days before beginning the tests. After transport to the behavioural room, mice were habituated for 1 h before any test. The investigators were blinded to group allocation during experiments.

For the NOR and novel object location tests, mice were tracked with an overhead FireWire camera (DMK 31AF03-Z2, The Imaging Source) and ANY-maze (Stoelting). All apparatuses and testing chambers were cleaned with 70% ethanol wipes between each animal.

*NOR test.* The day after habituation to the grey acrylic arena (44 × 44 cm), mice were exposed to two identical objects for 10 min during the familiarization phase. Object preference was evaluated during this session. Testing occurred 1 and 24 h later in the same arena, to assess short-term memory and long-term memory, respectively. Mice were allowed to explore for 10 min the same arena but one of the familiar objects was changed by a new one. The objects used were different in shape, colour, size and material. Moreover, the position of the objects in the familiarization and test session was counterbalanced between animals. Any investigative behaviour towards (that is, head orientation, sniffing occurring within < 1.0 cm) or deliberate contact with an object was used as measure of the exploration and was registered manually by an experimenter blind to genotype and treatment. The exploration time for the Familiar object (Fam or F) and the new object (New or N) during the test phase was recorded. Memory was operationally defined by the percentage of DI for the novel object as the time spent investigating the new object minus the time spent investigating the familiar one in the testing period ($\%DI = ((NO - FO)/\text{Total Exploration Time}) \times 100$).

*SOL test.* The SOL test evaluates spatial memory by measuring the ability of mice to recognize the new location of a familiar object on the basis of the available extra-maze cues. After habituation, mice were exposed to two identical objects for 10 min during the familiarization phase. Object preference was evaluated during this session. Testing occurred 1 h later in the same arena. Mice were allowed to explore for 10 min the same arena but one of the familiar objects was moved to a novel location. Any investigative behaviour towards (that is, head orientation, sniffing occurring within < 1.0 cm) or deliberate contact with an object was used as the measure of exploration that was registered manually by an experimenter blind to genotype and treatment. The exploration time for the object in the old location (Old or O) and the object in the new location (New or N) during the test phase was recorded. Memory was operationally defined by the percentage of alternation index for the novel located object as the time spent investigating the object in the new location minus the time spent investigating the object in the old location in the testing period ($\%\text{Alternation} = ((NL - OL)/\text{Total Exploration Time}) \times 100$).

Each animal was tested in the novel object tasks only once. All experimental groups are thus fully independent. To exclude the possibility that NOR performance might be confounded by *a priori* spatial or object biases, placement of the novel object was alternated between the left and right corners of the open-field arena.

**Determination of GABA levels in CSF by LC–MS/MS.** Mouse CSF samples were collected from the cisterna magna following a previously described protocol[65].

GABA was quantified by ultra performance LC–MS/MS, following the protocol described by Buck et al.[41]. Briefly, GABA was extracted from CSF by precipitation with acetonitrile spiked with deuterated GABA (D6) as internal standard (Sigma Aldrich). Given the intrinsic difficulty of collecting a sufficient amount of CSF from mice, the CSF content of two or three individual animals were pooled into final samples consisting of 7 to 12 µl of CSF. A total number of 14 controls and 10 *Bmal1*cKO animals were used for this experiment (pooled into 5 and 4 samples, respectively). GABA was then separated by hydrophilic interaction LC using a BEH HILIC 2.1 × 100 mm column and a short gradient of water in acetonitrile (5 to 40% in 2 min), with both eluents added with formic acid to a final 0.1% v/v. Flow rate was kept at 0.45 ml min$^{-1}$. GABA was quantified on a Xevo TQ-MS instrument operating in electrospray, positive ion mode and following the multiple reaction monitoring transitions as previously described[41]. Both the column and the ultra performance LC–MS/MS systems were purchased from Waters, Inc. (Milford, MA, USA). GABA was quantified using a standard calibration curve prepared by serial dilution in artificial CSF[66].

The investigators were blinded to group allocation during experiments.

**Immunofluorescence.** Mice were administered ketamine/xylazine (150, 10 mg kg$^{-1}$) and transcardially perfused with ice-cold PBS followed by ice-cold 4% paraformaldehyde in PBS. Brains were post-fixed overnight in 4% paraformaldehyde in PBS and 30 µm slices were prepared Cryostat (Leica). Slices were permeabilized with 0.3% Triton X-100 in PBS, blocked with 10% goat serum in PBS and incubated at 4 °C overnight with the primary antibody (rabbit anti-VIP, 1:2,500 dilution (ab4384, Abcam); mouse anti-GFAP 1:1,000 dilution (Sigma, G3893); rabbit anit-Bmal1, dilution 1:200 (Abcam, ab93806); mouse anti-S100β, 1:1,000 dilution (Sigma, AMAB91038); rabbit anti-GAT3, 1:200 dilution (Abcam, ab431); and rabbit anti-GAT1, 1:200 dilution (Abcam, ab426)).

The following day, sections were extensively washed and incubated for 2 h with goat anti-rabbit or anti-mouse Alexa-488 or Alexa-546 secondary antibodies used at 1:1,000 dilution (Invitrogen, A-11034, A32723, A-11003 and A-11010). Slices were then washed, mounted with Prolong Gold and imaged in an inverted laser scanning confocal microscope (TCS SP5 microscope using a × 20 or × 40 objective (Leica Microsystems)). Quantification and analysis was performed in ImageJ software (Wayne Rasband, NIH, USA), by outlining the SCN or cortex from the 4,6-diamidino-2-phenylindole-stained image and using this template to measure the relative intensity of the immunostaining for VIP, GAT1 or GAT3. When more than one section was analysed from each animal, the mean of the measures from consecutive sections were used for that individual.

**Cell cultures and transfections.** Primary monolayer cultures of astrocytes or neurons were established from cerebral cortices of neonatal (P1–P3) or Embryonic day 17 (E17), respectively, Sprague–Dawley rats and maintained at 37 °C in a humidified atmosphere of 5% $CO_2$. Astrocyte cultures were prepared and maintained as previously described[67,68]. The cultures were maintained at 37 °C in a humidified atmosphere of 5% $CO_2$ for 1 week and thereafter cells were trypsinized and subcultured for the different experiments. These astrocyte-enriched cultures contained > 96% astrocytes as indicated by immunofluorescence, with a monoclonal antibody anti-GFAP (Clone 6F2, Dako; dilution 1:1,000).

For primary neuronal cultures, tissue was dissociated by digestion with 10% (v/v) Trypsin for 30 min at 37 °C and 200,000 cells were plated onto 20 mm tissue culture dishes coated with 0.1 mg ml$^{-1}$ poly-D-lysine. Neurons were grown in Neurobasal medium supplemented with B27, Gmax, 2.5 U ml$^{-1}$ Penicillin and 2.5 µg ml$^{-1}$ Streptomycin. Three days after plating, and subsequently every 4–5 days, half of the medium was changed. Neuronal cultures were maintained for up to 25 days *in vitro* before being used for the different experiments.

Astrocytes were synchronized with 100 nM of Dexamethasone for 2 h. The hormone was washed out and used for co-cultured experiments or harvested at different time points for subsequent analyses. Neurons were synchronized with 100 nm of Dexamethasone, GABA (100 µM), glutamate (10 µM) or with vehicle in neuronal conditional medium for 2 h. For the co-culture experiments, astrocytes were plated on coverslips on 12-well dishes and grown until confluence. Cortical neurons were plated on coated 12-well dishes with paraffin feets, to avoid contact between the cultures. Astrocyte grown on coverslips were inverted and resting on the dish containing the neurons sharing the same culture media, and harvested at different time points for subsequent analysis. 1(S), 9(R)-(−)-Bicuculline methiodide was diluted in cultured medium (stock 10 mg ml$^{-1}$) and cells were treated with 30 µM. Cells treated with vehicle were used as controls.

Primary astrocytes were transiently transfected with Bmal1 siRNAs (ON-TARGET plus smartpool specific for rat *Bmal1*, Dharmacon) or scramble control (ON-TARGET plus non-targeting siRNAs, Dharmacon). Briefly, primary astrocytes were plated on coverslips on 12-well dishes and when reached 60–80% confluence were transfected with 10 nM of scramble control or Bmal1 siRNAs using the Lipofectamine RNAiMAX Transfection Reagent (Thermo Fisher Scientific) following the suggestions of the manufacturer. Medium was changed 24 h later and on the next day (48 h after transfection) were subjected for consequent experiments.

**RNA isolation and quantitative real-time RT–PCR.** Cells were harvested at the appropriate time points, with the time of treatment being defined as CT4. For each time point, we prepared samples for the assay in triplicate. Tissues (cortices and hippocampus) were collected for RNA isolation at the ZTs of interest from animals in LD cycles (12–12 h). Total RNA was extracted using TRIzol reagent following the manufacturer's instructions. RNA was further cleaned using an RNeasy Mini Kit. Complementary DNA was obtained by retrotranscription of 0.5 μg of total mRNA using the ImProm-II Reverse Transcription System following the manufacturer's instructions. Real-time reverse transcriptase–PCR was done using the ABI PRISM.7900 (Applied Biosystems). For a 15 μl reaction, 9 ng of cDNA template was mixed with the primers to a final concentration of 200 nM and mixed with 7.5 μl of 2× QuantiFast SYBR Green PCR Master Mix. The reactions were done in duplicates using the following conditions: 5 min at 95 °C followed by 40 cycles of 10 s at 95 °C, 30 s at 60 °C and 1 min at 70 °C. The primers used are available on request. *Gapdh* or *Pgk1* transcripts were used as reference controls.

**Western blotting.** Cortices were homogenized in lysis buffer with the following final concentrations: 50 mM Tris-HCl pH 7.5, 250 mM sucrose, 5 mM sodium pyrophosphate (NaPPi), 50 mM NaF, 1 mM EDTA, 1 mM EGTA, 1 mM dithiothreitol, 0.5 mM phenylmethylsulfonyl fluoride, 0.1 mM benzamidine, 50 μg ml$^{-1}$ leupeptin and 50 μg ml$^{-1}$ soybean trypsin inhibitor. Following lysis, SDS was added to a final of 0.2%. Cell pellets were lysated by heating at 95 °C for 5 min in 1% SDS and immediately cooled at 4 °C for 15 min with ice-cold lysis buffer (50 mM Hepes pH 7.5, 150 mM NaCl, 10% Glycerol, 1% Triton X-100, 5 mM EGTA, 1.5 mM MgCl$_2$, 20 mM Na$_4$P$_2$O$_7$; 20 mM Na$_3$VO$_4$; 50 μg ml$^{-1}$ aprotinin and 4 mM phenylmethylsulfonyl fluoride). After centrifugation (15,000 g, 15 min, 4 °C) to separate cellular debris, the lysates were resolved in a 10% SDS–PAGE and electrotransferred onto a nitrocellulose membrane. Membranes were then probed with antibodies against BMAL1 (Abcam, ab93806), CRY1 (Thermo Fisher, PA5-13124) and PER2 (Santa Cruz, H-90: sc-25363) (dilution 1:1,000), and extensively washed and re-probed with antibodies against glyceraldehyde 3-phosphate dehydrogenase (Santa Cruz, FL-335: sc-25778 dilution 1:1,000) as loading control. Horseradish peroxidase-coupled secondary antibodies were purchased from Abcam and used at 1:5,000 dilution (ab97046 and ab6721). Immunoreactive bands were detected with a westernlight chemiluminescence detection system (ECL, GE Healthcare Bio-Sciences AB), exposed and photographed in an ImageQuant LAS 4,000 mini (GE Healthcare Bio-Sciences AB). Images have been cropped for presentation. Full-size images are presented in Supplementary Figs 5 and 7.

**GABA uptake assay.** Primary astrocytes were grown and transfected in 12-multiwell plates with *Bmal1* and Scrbl siRNAs. After 48 h, cells were rinsed once with PBS and pre-incubated with 1 ml of Hank's buffered salinesolution for 10 min at room temperature. GABA was added to the medium to a final concentration of 5, 10 and 40 μM, and the cells were incubated for 15 min at room temperature. The medium was collected and GABA levels analysed with the GABA research ELISA kit (Labor Diagnostica Nord GmbH & Co), following the manufacturer's instructions.

**Statistical analysis.** Data are presented as mean ± s.e.m. and were analysed and graphed using Prism 6 (GraphPad, San Jose, CA, USA). Statistical comparison of two groups was done by Student's two-tailed unpaired *t*-test or two-way analysis of variance with a *post-hoc* Bonferroni. Data were checked for normality and equal variances between groups. $P < 0.05$ was considered as statistically significant and the significance is marked by *$P < 0.05$, **$P < 0.01$ and ***$P < 0.001$. Statistical significance of the rhythmic expression was determined by Cosinor analysis (expression data was fit by a nonlinear least-squares regression with the following equation: $y = A + B \times \cos[2\pi \times (t - C)/24]$, where $A$ is the rhythm-adjusted mean, $B$ is the amplitude of the rhythm, $C$ is the phase given in circadian time representing the time of peak expression and $t$ is the circadian time). The required sample size was calculated based on the similar experiments and analyses carried out previously. The number of animals in each experiment is stated in the respective figure legends. Samples or animals were excluded from the data analysis with pre-established criteria, if they deviated more than 2 s.d. from the group mean.

**Data availability.** All raw/original relevant data are available upon request.

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

## Acknowledgements

We thank Dr M. Götz (Physiological Genomics, Biomedical Center, Ludwig-Max-imilians-University Munich, Germany) and Dr S.J. Moore (Molecular and Behavioral Neuroscience Institute, University of Michigan, Ann Arbor, Michigan, USA) for kindly providing the Glast-Creert2 and Bmalflox/flox mouse lines, respectively. We thank Tecniplast Spa (Buguggiate, Italy) for providing individually ventilated cages. We thank Drs T. Fellin and A. Barberis (IIT-NBT, Genova, Italy) for critical review of the manuscript and Dr I. Fernandez (IIT-NBT) for advice and discussion. We thank R. Pelizzoli and IIT-NBT technical staff (M. Pesce, C. Chiabrera, F. Succol and M. Nanni) for their excellent support. We also thank the Animal Facility of IIT central research labs in Genoa (F. Piccardi, D. Cantatore, R. Navone and M. Morini) for assistance in animal experiments. D.D.P.T. and L.B. were supported by intramural funds of Fondazione Istituto Italiano di Tecnologia (IIT). O.B.-M. was supported by the European Research Executive Agency (REA) through the FP7-PEOPLE-2014-IEF 'ASTROCLOCK' (629867).

## Author contributions

O.B.-M. designed and performed the experiments, analysed the data and wrote the manuscript. M.P.-E. performed and analysed the behavioural experiments. P.F. quanti-fied the immunofluorescence experiments, A.A. quantified and analysed the levels of GABA in CSF. D.D.P.T and L.B. conceived and coordinated the project, edited and revised the manuscript. All authors approved the final version of the manuscript.

## Additional information

**Competing financial interests:** The authors declare no competing financial interests.

**Publisher's note**: 

