## [Peer Review File · Nature Communications]

Reviewers' comments:

Reviewer #1 (Remarks to the Author):

This is a very novel, interesting and potentially exciting study. The authors provide strong data indicating that astrocytic BMAL plays a critical role in controlling clock neurons, and rhythms in clock components outside the SCN. They also find parallels with their results and earlier results indicating that SCN outputs influence cognition. I have some comments that if addressed would strengthen the paper.

1) Figure 1 lacks sufficient IHC data to conclude that Bmal1 knockdown is specific to astrocytes. Why did the investigators use a GFAP antibody to identify astrocytes when Cre expression is driven by a GLAST promoter? In doing so, a large number of Cre-expressing cells are not associated with a cell type. In addition for staining for GLAST, the paper would be greatly improved if it included additional staining for other cell types including neurons. DAPI stains cell nuclei, but by itself, does not discriminate between neurons or other types of glia.

2) It is not clear if the cognitive phenotype is really due to deletion of astrocyte Bmal1 in SCN astrocytes, as suggested by the authors. This is because a global knockdown strategy is used. It is also possible that knockdown in cortical or hippocampal astrocytes is the cause. In order to make the claim that this result is due to SCN deletion, they have to do a SCN specific knockdown (e.g using a viral vector approach). Likewise, systemic PTZ or PTX treatments are used to rescue the wt phenotype in the KO; however, in order to conclude this reflects action in the SCN and not elsewhere, a targeted injection/infusion would be needed.

3) There appears to be a general reduction in ALL Bmal+ cells in the SCN following the tamoxifen treatment (Fig 3). There is no IHC performed to determine how much of this reduction is neuronal or astrocytic.

4) Figure 1 shows that GFAP staining itself appears to be greatly reduced in the ventral portion of the SCN in the KO-but DAPI staining seems unchanged. This means that GFAP+ astrocyte cell numbers are also being reduced by the tamoxifen treatment. This needs to be investigated and if true, the results need to re-examined in light of the loss of GFAP+ cells.

5) Please indicate if control mice were also treated with tamoxifen. If not, why not?

6) The text states that mice were in LD for 10 days, DD for 30 days, and LD for another 10 days. However the actograms in Figure 2 seem to show 8 days LD, 28 days DD, and 8 days LD. There also appears to be data missing for rLD day 6 for the Bmal1 cKO actogram. Can the authors comment on these issues?

7) This statement is confusing "However, in constant darkness, Bmal1cKO mice showed a bimodal pattern of locomotor activity and a significant phase-shift advance in the rhythm of activity of 1.31 h ($p < 0.001$, two-tail t-test)". Neither the actograms nor the corresponding activity plots show a phase advance in DD. Are the authors referring to the first mode in the bimodal distribution in the KOs under DD? Or the activity in rLD, which does appear to be phase advanced in the KOs. Please clarify.

8) What was the rationale for using rats for the culture experiments?

9) The legend provided for the neuronal graphs in Figure 4b are confusing. The labeling suggests that neurons were also treated with dexamethasone and siRNA.

10) What was the LD/DD protocol used in Figure 6? It seems to differ from Figure 2 in that 1) there's no return to LD, and 2) 15 days were spent in LD and 24 days in DD in Figure 6b. What was the rationale for differing protocols?

11) Figure 6b could benefit from direct statistical comparisons between control and Bmal1 cKO as in Figure 2. Furthermore, direct statistical comparisons of activity between Bmal1 cKO mice in Figure 2 (or Bmal1 cKO mice that received control injections for PTZ if that experiment was done) and Bmal1 cKO treated with PTZ from Figure 6 could be made.

12) Please clarify what "untreated animals" means for the definitions of statistical markers in the Figure 6 legend.

13) Paragraph in the main text that refers to Fig 4 has call-outs that are confusing.

Reviewer #2 (Remarks to the Author):

In this manuscript the authors investigated the effects of a deletion of the clock gene *Bmal1* in a subpopulation of astrocytes (*Glast* expressing astrocytes). They find that the mutated animals display altered re-adaptation to the light dark cycle after constant darkness, indicating abnormal masking in these mice. Furthermore altered behavior in the novel object recognition test was observed. At the molecular level the mutation appears to lower the amplitude of clock gene expression in the cortex and hippocampus. Interestingly, *Vip* levels in the SCN were increased. In vitro *Bmal1* knockdown astrocytes suppress entrainment of co-cultured cortical neurons supporting the hypothesis that in the astrocyte *Bmal1* ko animals coordination of the clock in the SCN is hampered. For the signaling in the co-culture experiment GABAA receptors appear to be important and antagonists for this receptor rescue the behavioral phenotype of *Bmal1* astrocyte KO mice.

Although it becomes clear after reading the article that only *Glast* positive astrocytes lack *Bmal1*, it should be stated at least in one of the subtitles. Instead of Astrocyte specific deletion of *Bmal1* .. it would be more precise to formulate it as Deletion of *Bmal1* in *glast*-positive Astrocytes.

The in vitro assay suggests an alteration in GABA signaling. To take this observation to the organismal level, the authors should measure whether the GABA levels in the mutated mice are indeed altered in the extracellular space of the brain.

In the experiment to figure 5 it would be advisable to use a substance unrelated to GABA signaling as control. Since *Bmal1* is deleted in *Glast*-positive astrocytes a control for glutamate levels is important, since absence of *Bmal1* may affect the expression of *Glast* in these astrocytes. Hence glutamate signaling may be altered and not only GABA signaling.

In the experiment to figure 6 GABA rescues the phenotype of activity distribution, however the authors did not look at the main phenotype, the re-adaptation to the new light-dark cycle. If this is not rescued by GABAA antagonists, this phenotype may be related to changes in Glutamate signaling. Therefore this experiment has to include the transition from DD to LD. Whatever the result is it will tell for which part of the locomotor phenotype GABA signaling is important and for which part Glutamate signaling is important. I should point out that for light mediated resetting of the circadian clock glutamate signaling is important, which is the reason why glutamate signaling should be analyzed in these animals.

Minor points:

At times the article is difficult to read. In the second paragraph of the discussion I could not understand the logic which may be due to language reasons. Therefore I would suggest to improve the wording of the article.

Figure 1: the P value for *** is given but in d only ** are there.

Figure 3: correct *Gadph* to *Gapdh* (it is glyceraldehyde 3-phosphate dehydrogenase)

Figure 4: the different shades of blue and red are hard to discriminate.

Reviewer #3 (Remarks to the Author):

Recent work has supported the view that astrocytes are together with neurons important components of functional neural networks. Dr. Barca-Mayo and colleagues examined the role that astrocytes play

in the generation of circadian rhythms by the suprachiasmatic nucleus. They used a conditional *Glast-Cre, Bmal1^{flx/flx}* expressing mice to selectively knock down *Bmal1* in a subset of astrocytes and examine how disruption of this circadian clock gene in astrocytes would affect behavioral and gene expression circadian rhythms. The results provide strong evidence for a role of *Bmal1* expression in cortical and hippocampal neurons altering the performance on behavioral tasks. A major limitation of the manuscript is that the conclusions and descriptions of the behavioral data do not match the data shown.

Major concerns:

1. In Figure 1a there are three populations, GFAP-expressing astrocytes, Tomato-expressing astrocytes and those that express both GFAP and Tomato. Also, *BMAL1* was expressed in only 43.7% of GFAP-expressing cells and reduced in ~60% of this subset of cells so that ~18% of astrocytes continue to express *Bmal1*. These data raise a concern about whether the identity of the specific cells with knocked-out *Bmal1* can be determined. Why is *Bmal1* not reduced in all *Glast-Cre* cells? Does *Glast-CreERT2* drive recombination in *S100-beta*-positive or glutamate synthase-positive astrocytes in the SCN.
2. The description of the running wheel activity does not match the data shown in Figure 2. The authors indicate that there is a 1.31 hour phase advance of the wheel running activity in the *Bmal1cKO* mice. There is no phase advance apparent in the data shown in Fig. 2b. The blue lines in the average activity graphs show the onset of activity in control and *Bmal1cKO* mice to be very similar or possibly the activity in the *Bmal1cKO* mice start a little later, which would indicate a altered phase angle of entrainment not a phase change.
3. The rate of re-entrainment is not directly studied and the data presented provide no support for the claim that the rate of re-entrainment is affected.
4. Given the small effects of the *Bmal1* knock-out on the locomotor activity the in vivo effects of the picrotoxin or PTZ injections are hard to interpret because they may just reflect the variability of the running wheel behavior of individual mice.
5. No experiments were performed nor data shown that is consistent with the statement "suggesting a defect in their masking responses".
6. *Bmal1* is a canonical clock gene and an important transcription factor. One of the challenges in studying *Bmal1* knock-outs is to distinguish which observed effects are due to circadian clock disruption and which are due to disruption of gene transcription. The effects on learning and memory may result in a disruption of astrocyte signaling in the cortex and hippocampus that is independent of any disruption of a circadian clock in the astrocytes.
7. The behavioral data should be analyzed with a periodogram to clearly demonstrate the presence of the bimodal behavior.

Minor concerns:

1. The meaning of the following sentence is unclear. "During the light-dark condition, *Bmal1cKO* mice showed similar robust circadian locomotor activity than control mice (Fig. 2b), with no differences in the periodicity, neither in the total average activity (Fig. 2c)."
2. "In contrast, *VIP* expression was not downregulated in the SCN of *Bmal1cKO* mice at ZT12 (Fig. 3b) suggesting constitutive repression of *VIP* expression by astrocytic *Bmal1* leading to impairment of circadian rhythmicity of *VIP* in the mutant mice." I suggest taking constitutive out of this sentence

since the Bmal1 expression is rhythmic and the data suggest that the Bmal1 rhythm contributes to the VIP rhythm.

3. On pg. 3 - reference 28 is not the correct reference.

4. Reference 18 demonstrates a strong impairment on the novel recognition test in Bmal1 knockout mice.

Reviewer #1 (Remarks to the Author):

This is a very novel, interesting and potentially exciting study. The authors provide strong data indicating that astrocytic BMAL plays a critical role in controlling clock neurons, and rhythms in clock components outside the SCN. They also find parallels with their results and earlier results indicating that SCN outputs influence cognition. I have some comments that if addressed would strengthen the paper.

Authors' Answer. We thank the reviewer for positive comments and strong support of this study. In particular we are grateful that the reviewer acknowledges that *"This is a very novel, interesting and potentially exciting study"*. **We have addressed your comments by performing new experiments and by revising the MS.** The revised text is highlighted in red in the MS. The point-by-point response follows. We sincerely hope that we have addressed all your concerns appropriately.

Reviewer's specific comment 1) *Figure 1 lacks sufficient IHC data to conclude that Bmal1 knockdown is specific to astrocytes. Why did the investigators use a GFAP antibody to identify astrocytes when Cre expression is driven by a GLAST promoter? In doing so, a large number of Cre-expressing cells are not associated with a cell type. In addition for staining for GLAST, the paper would be greatly improved if it included additional staining for other cell types including neurons. DAPI stains cell nuclei, but by itself, does not discriminate between neurons or other types of glia.*

Authors' Answer. As suggested, we have now quantified the co-localization of Glast-Cre-driven Td-tomato reporter with the astrocyte markers GFAP and S100 β (new Fig. 1 and supplementary Fig. 1 of the revised MS). In particular, we found that approximately 60% of Td-Tomato positive cells were expressing GFAP and S100 β , thus confirming that Glast-CreERT2 drives recombination in SCN astrocytes.

Moreover, we have quantified the percentage of BMAL1 positive cells in Glast-Cre-ER^{T2} +/-, Td-Tomato positive cells in the SCN of both Bmal1cKO and control mice over the total Tomato positive, total S100 β positive, or over total GFAP positive astrocytes. We found a significant reduction (that was ranging between 70-60% depending on the marker used for quantification) in the proportion of BMAL1 positive astrocytes (new Fig1 and supplementary Fig1 of the revised MS). This result demonstrates that depletion of Bmal1 occurs in the majority of the SCN astrocytes in our mutant mice.

We have also tried to trace the expression of Td-Tomato in SCN neurons, by using NeuN but we found a very limited expression of this pan-neuronal marker (see figure 1a for referees enclosed). This result is consistent with previous observations indicating lack of NeuN expression in SCN neurons (Moldavan, M., Cravetchi, O., Williams, M., Irwin, R.P., Aicher, S.A., Allen, C.N. Localization and expression of GABA transporters in the suprachiasmatic nucleus. *Eur J Neurosci.* 42, 3018-32 (2015).

Remarkably, in the mutant mice, BMAL1 expression was significantly reduced also in Td-Tomato negative cells (new Fig. 1C). Consistently, we found that the fluorescence intensity of BMAL1 in the SCN was reduced by 78.9 % ($p=0.0003$, two-tail T-test) in Bmal1cKO, compared to control mice. Therefore, this result indicates that astrocyte-specific deletion of Bmal1 results in a global down-regulation of BMAL1 protein in the SCN, in agreement with our hypothesis that deletion of Bmal1 in astrocytes might affect the neuronal clock.

Reviewer's specific comment 2) *It is not clear if the cognitive phenotype is really due to deletion of astrocyte Bmal1 in SCN astrocytes, as suggested by the authors. This is because a global knockdown strategy is used. It is also possible that knockdown in cortical or hippocampal astrocytes is the cause. In order to make the claim that this result is due to SCN deletion, they have to do a SCN specific knockdown (e.g using a viral vector approach). Likewise, systemic PTZ or PTX treatments are used to rescue the wt phenotype in the KO; however, in order to conclude this reflects action in the SCN and not elsewhere, a targeted injection/infusion would be needed.*

Authors' Answer. We agree with this comment. However, we respectfully remark that the aim of our study is not to define whether the alterations of the cognitive phenotype of Bmal1cKO are due to the deletion of Bmal1 in astrocytes of the SCN vs other brain regions. Indeed, we have specified in the

"Introduction" on page 3: "The SCN, as other brain regions, are composed of a heterogeneous population of cells, including astrocytes...recent evidence suggests an involvement of astrocytes in the regulation of circadian rhythms in *Drosophila*^{5,6} and in mammals^{7,8}, the role of clock genes in these cells has not been investigated.It is currently unknown whether the regulation of astrocyte physiology by clock genes might contribute to the maintenance of neuronal rhythmic behaviour at cellular, tissue and organism level...Identifying such a role for astrocytes clock genes will not only reveal a more complex cellular signalling in brain than the considered so far..."", and in "Results" on page 5: "Altogether, our results indicate that the selective ablation of *Bmal1* in adult astrocytes is sufficient to alter daily locomotor activity and declarative memory in mice. These phenotypes might be dependent on astrocytic *Bmal1* functions affecting rhythmic oscillations in the SCN and/or in cortical and hippocampal circuits involved in memory".

Reviewer's specific comment 3) *There appears to be a general reduction in ALL Bmal+ cells in the SCN following the tamoxifen treatment (Fig 3). There is no IHC performed to determine how much of this reduction is neuronal or astrocytic.*

Authors' Answer. Thanks for this suggestion. We have performed new experiments to answer this question. Data are now shown in Fig.1, Supplementary Fig. 1 of the revised MS, and Fig.1 for reviewers enclosed in this document. Please, see also answer to specific comment 1.

Reviewer's specific comment 4) *Figure 1 shows that GFAP staining itself appears to be greatly reduced in the ventral portion of the SCN in the KO-but DAPI staining seems unchanged. This means that GFAP+ astrocyte cell numbers are also being reduced by the tamoxifen treatment. This needs to be investigated and if true, the results need to re-examined in light of the loss of GFAP+ cells.*

Authors' Answer. We have quantified the percentage of GFAP and Td-Tomato positive cells in the SCN of control and *Bmal1*cKO-Td-Tomato animals and found no differences among the two genotypes (see Fig.1b for reviewers attached). We have added a better representative image for *Bmal1*cKO mice in the new supplementary Fig. 1b of the revised MS showing the presence of GFAP positive astrocytes in the ventral SCN of the mutants.

Reviewer's specific comment 5) *Please indicate if control mice were also treated with tamoxifen. If not, why not?*

Authors' Answer. All animals, including controls, were treated with tamoxifen, as indicated in "Methods" in the section "Animals and treatments": "6-8 weeks old *Glast-CreERT2; Bmal1*^{flx/flx} (*Bmal1*cKO) and controls (*Bmal1*^{flx/flx}) were treated with tamoxifen dissolved in corn oil"

Reviewer's specific comment 6) *The text states that mice were in LD for 10 days, DD for 30 days, and LD for another 10 days. However the actograms in Figure 2 seem to show 8 days LD, 28 days DD, and 8 days LD. There also appears to be data missing for rLD day 6 for the Bmal1 cKO actogram. Can the authors comment on these issues?*

Authors' Answer. We thank the reviewer for noticing this point that allowed us to improve the clarity of our MS. Mice were subjected to wheel-running activity assay on different rounds and there might be small variation in the number of days in each lighting condition on the different sets of experiments. In the revised MS (see Results and Methods sections) we now specify that "mice were kept in LD for at least 8 days, followed by 3-4 weeks on DD and at least 8 days in rLD". Since we keep the mice in DD for almost one month, this small variations in the days are not under- or overestimating the phenotype of the mice.

We have also verified the recording on rLD at day 6 for the representative *Bmal1*cKO mouse reported in the Fig. 2, and found very low levels of activity at that day, thus indicating that there was no gap in our recording.

Reviewer's specific comment 7) *This statement is confusing "However, in constant darkness, Bmal1cKO mice showed a bimodal pattern of locomotor activity and a significant phase-shift advance*

in the rhythm of activity of 1.31 h ($p < 0.001$, two-tail t-test)". Neither the actograms nor the corresponding activity plots show a phase advance in DD. Are the authors referring to the first mode in the bimodal distribution in the KOs under DD? Or the activity in rLD, which does appear to be phase advanced in the KOs. Please clarify.

Authors' Answer. We thank the reviewer for this comment and we agree that this sentence might be misunderstood. In the revised MS, we have modified the representation of the total average activity (new Fig. 2c) and added: a) the periodogram to show the bimodal pattern of activity (new Fig 2b, right panels); b) the activity onset in DD showing that Bmal1cKO mice delayed their active phase as compared to control animals (new Fig 2d, left panel); c) onset and offset of activity in rLD, showing that Bmal1cKO mice advanced the onset and offset in the wheel running activity as compared to control animals (new Fig 2d, middle and right panels). Accordingly, we also amended the text in relevant parts.

Reviewer's specific comment 8) *What was the rationale for using rats for the culture experiments?*

Authors' Answer. Although we were able to synchronize mouse cortical neurons and astrocytes by using the dexamethasone protocol, the co-culture experiments were difficult to reproduce with mouse cells. Thus, we used the rat model for the in vitro co-culture experiments since mimic the results obtained in our in vivo model. This suggests that the mechanism of astrocyte to neuron circadian communication is conserved among those two species.

Reviewer's specific comment 9) *The legend provided for the neuronal graphs in Figure 4b are confusing. The labelling suggests that neurons were also treated with dexamethasone and siRNA.*

Authors' Answer. We have modified the legend of the Fig. 4b and in the corresponding text of the new version of the MS.

Reviewer's specific comment 10) *What was the LD/DD protocol used in Figure 6? It seems to differ from Figure 2 in that 1) there's no return to LD, and 2) 15 days were spent in LD and 24 days in DD in Figure 6b. What was the rationale for differing protocols?*

Authors' Answer. We have now added the data on the re-entrainment to the LD (rLD) and also performed new experiments to increase the number of animals subjected to the circadian locomotor behaviour after treatment with PTZ. We found no differences in the results obtained in DD (i.e. in the periodicity, periodogram, onset and offset of activity) between the animals that stayed 3 or 4 weeks in constant darkness. Accordingly, we have now specified in the text (Results and Methods sections) of the revised MS that: "mice were in LD for at least 8 days, 3-4 weeks on DD and at least 8 days in rLD".

Reviewer's specific comment 11) *Figure 6b could benefit from direct statistical comparisons between control and Bmal1 cKO as in Figure 2. Furthermore, direct statistical comparisons of activity between Bmal1 cKO mice in Figure 2 (or Bmal1 cKO mice that received control injections for PTZ if that experiment was done) and Bmal1 cKO treated with PTZ from Figure 6 could be made.*

Authors' Answer. As suggested by the reviewer, we have added in the revised MS (new Fig. 7) the comparison among Bmal1cKO and PTZ-treated Bmal1cKO in DD cycles, as well as direct statistical comparisons between PTZ treated control and PTZ-treated Bmal1cKO animals in DD (new Fig 7).

However, we would like to remark that our aim was to compare control and Bmal1cKO mice upon treatment with the GABA antagonist. Indeed, many studies do not include the group of "control treated animals" when the aim is not to test the effects of the drug itself but the effects of the genotype, specially when there might be a significant interaction between genotype and drug treatment. Indeed, PTZ treatment in control animals shift the phase of activity although the mechanism is unknown. We have observed a similar trend in our control and Bmal1cKO mice in LD cycles. Remarkably, no differences were found among genotypes suggesting a direct effect of the drug and not of the genotype. On the other hand, the protocol treatment (at ZT6) that we follow, previously described by Ruby and Colas, might be the cause of this shift due to the arousal effect of

the injections, which is known to stimulate activity when injections are given repeatedly. Consistently, it is common knowledge in the food entrainment literature, that such activity pattern can be observed long after the scheduled feeding is terminated. Again we argue that, as supported by our data, this effect is genotype-independent.

Reviewer's specific comment 12) Please clarify what "untreated animals" means for the definitions of statistical markers in the Figure 6 legend.

Authors' Answer. Thank you for pointing this out. We have clarified that "untreated animals" refers to animals "not treated neither with PTZ nor PTX" see the legend of the new Fig. 7 in the revised MS.

Reviewer's specific comment 13) Paragraph in the main text that refers to Fig 4 has call-outs that are confusing.

Authors' Answer. We have modified the call-outs of the Fig. 4 in the new MS

We are grateful for the constructive criticism and helpful suggestions. We sincerely hope you will consider the revised manuscript as ready for publication in Nature Communications now.

Fig1. (a) Representative micrographs of NeuN (red) and Dapi (blue) immunostaining, showing a very limited expression of NeuN in the SCN. **(b)** Quantification of the Td-Tomato positive (left panel) and GFAP positive (right panel) in the SCN of control and Bmal1cKO animals, showing no differences among genotypes.

Reviewer #2 (Remarks to the Author):

In this manuscript the authors investigated the effects of a deletion of the clock gene Bmal1 in a subpopulation of astrocytes (Glast expressing astrocytes). They find that the mutated animals display altered re-adaptation to the light dark cycle after constant darkness, indicating abnormal masking in these mice. Furthermore altered behavior in the novel object recognition test was observed. At the molecular level the mutation appears to lower the amplitude of clock gene expression in the cortex and hippocampus. Interestingly, Vip levels in the SCN were increased. In vitro Bmal1 knockdown astrocytes suppress entrainment of co-cultured cortical neurons supporting the hypothesis that in the astrocyte Bmal1 ko animals coordination of the clock in the SCN is hampered. For the signaling in the co-culture experiment GABAA receptors appear to be important and antagonists for this receptor rescue the behavioral phenotype of Bmal1 astrocyte KO mice.

Authors' Answer. We thank the reviewer for constructive criticism, which helped us to improve the quality of our study. **We have addressed your comments by performing new experiments and by modifying the MS.** The revised text is highlighted in red in the MS. The point-by-point response follows. We sincerely hope that we have addressed all your concerns appropriately.

Reviewer's specific comment 1) *Although it becomes clear after reading the article that only Glast positive astrocytes lack Bmal1, it should be stated at least in one of the subtitles. Instead of Astrocyte specific deletion of Bmal1 it would be more precise to formulate it as Deletion of Bmal1 in Glast-positive Astrocytes.*

Authors' Answer. According to your suggestion, we have modified the subtitles of the Fig. 1 and 2 in the "Results" section.

Reviewer's specific comment 2) *The in vitro assay suggests an alteration in GABA signalling. To take this observation to the organismal level, the authors should measure whether the GABA levels in the mutated mice are indeed altered in the extracellular space of the brain.*

Authors' Answer. We agree with the logic of the referee. As suggested, we have measured the levels of GABA in the cerebrospinal fluid (CSF) of both control and Bmal1cKO mice (at ZT6, since it is the time at which we administer the GABAA receptor antagonist). We found that Bmal1cKO mice had significantly elevated GABA levels, as shown in the new Fig. 6d of the revised MS. Since there are several different molecular pathways and compartments for enrichment, synthesis, and degradation of GABA, the resulting concentration of GABA in synaptic vesicles and in the extracellular space might depend on the equilibrium between these mechanisms. Indeed, the absolute concentrations of GABA in the presynaptic cytosol, in vesicles, and in the extrasynaptic space are not known. However, the affinity constants of extrasynaptic GABA receptors may serve as a rough estimate of background concentrations (0.2–2.5 μM) (Glykys et al, 2007) and in fact, in rat, direct measurements from cerebrospinal fluid yielded similar or slightly higher values (Eckstein et al., 2008). Thus, the CSF levels of GABA might reflect the concentrations of this neurotransmitter at the extrasynaptic space.

J. Glykys and I. Mody, "The main source of ambient GABA responsible for tonic inhibition in the mouse hippocampus," *Journal of Physiology*, vol. 582, no. 3, pp. 1163–1178, 2007.

J. A. Eckstein, G. M. Ammerman, J. M. Reveles, and B. L. Ackermann, "Analysis of glutamine, glutamate, pyroglutamate, and GABA in cerebrospinal fluid using ion pairing HPLC with positive electrospray LC/MS/MS," *Journal of Neuroscience Methods*, vol. 171, no. 2, pp. 190–196, 2008.

Reviewer's specific comment 3) *In the experiment to figure 5 it would be advisable to use a substance unrelated to GABA signalling as control. Since Bmal1 is deleted in Glast-positive astrocytes a control for glutamate levels is important, since absence of Bmal1 may affect the expression of Glast in these astrocytes. Hence glutamate signalling may be altered and not only GABA signalling.*

Authors' Answer. We agree with the comment of the reviewer that absence of Bmal1 might affect the expression of Glast, thus leading to an alteration in glutamate signalling.

We analyzed the expression of glutamate transporters in cortex of our mutant mice as well as in astrocytes upon knockdown of Bmal1 in vitro. We found that the expression of Glast in cortices of

control and Bmal1cKO animals at different ZTs was significantly downregulated in the mutant animals (see Fig. 2a for referees enclosed). We also investigated a potential compensation in GLT1 expression and found that it was not upregulated in the mutant, in fact it was significantly reduced at ZT6 (see Fig. 2a for referees enclosed). On the other hand, Herzog found that astrocytic Glast expression is under control of Clock, NPAS and Per2 genes (Beaule et al., 2009). Consistent with this study, we observed that knockdown of Bmal1 in primary astrocyte cultures leads to a downregulation not only of Glast but also Glt1. Thus, it is difficult to distinguish in vivo, whether the downregulation of Glast might be due to our cre-deletor mouse model or to a direct control by loss of Bmal1. Remarkably, we also found that the treatment with GABAA receptor antagonist PTX rescued the circadian and cognitive phenotype without restoring the expression of Glast and Glt1 in the cortex of Bmal1cKO (see Fig. 2b referees). Indeed, we now used glutamate (10uM) to verify whether it could synchronize cortical neurons. We found that glutamate, in contrast to GABA, does not synchronize the neuronal clock in vitro (we have added this data in the new Fig. 5a and "Results" section of the revised MS).

In summary, given that glutamate does not synchronize neurons, and that GABAA receptor antagonist rescued the phenotype without restoring the levels of glutamate transporters in our mutant mice, we concluded that glutamate might not be the primary neurotransmitter involved in the phenotypes of Bmal1cKO.

Beaulé C, Swanstrom A, Leone MJ, Herzog ED. Circadian modulation of gene expression, but not glutamate uptake, in mouse and rat cortical astrocytes. PLoS One. 2009 Oct 15;4(10):e7476.

Reviewer's specific comment 4) *In the experiment to figure 6 GABA rescues the phenotype of activity distribution, however the authors did not look at the main phenotype, the re-adaptation to the new light-dark cycle. If this is not rescued by GABAA antagonists, this phenotype may be related to changes in Glutamate signaling. Therefore this experiment has to include the transition from DD to LD. Whatever the result is it will tell for which part of the locomotor phenotype GABA signaling is important and for which part Glutamate signaling is important. I should point out that for light mediated resetting of the circadian clock glutamate signaling is important, which is the reason why glutamate signaling should be analyzed in these animals.*

Authors' Answer. We agree with the logic of the reviewer. Following your suggestion, we have now repeated these experiments and added the data of the rLD of control and Bmal1cKO animals after PTZ treatment (new Fig. 7 of the revised MS). Our data shows that the GABAA antagonist, PTZ, rescued all the phenotypes of Bmal1cKO mice, including the re-adaptation to new light cycle, thus suggesting that glutamate might not be directly involved in the circadian locomotor behaviour of our mutants.

Reviewer's Minor points:

-At times the article is difficult to read. In the second paragraph of the discussion I could not understand the logic which may be due to language reasons. Therefore I would suggest to improve the wording of the article.

Authors' Answer. We have revised the wording in the MS

*-Figure 1: the P value for *** is given but in d only ** are there.*

Authors' Answer. We have added more data for the recombination of Bmal1 in SCN astrocytes (see new Fig. 1 and supplementary Fig. 1 or the revised MS) and modified the P values accordingly.

-Figure 3: correct Gadph to Gapdh (it is glyceraldehyde 3-phosphate dehydrogenase)

Authors' Answer. Ok

-Figure 4: the different shades of blue and red are hard to discriminate.

Authors' Answer. We have modified the colours of Fig. 4.

We are grateful for the constructive criticism and helpful suggestions. We sincerely hope you will agree that the revised manuscript is ready for publication in Nature Communications now.

[Unpublished data removed as per authorial request by Editorial Team for the Supplementary Peer Review File.]

Reviewer #3 (Remarks to the Author):

*Recent work has supported the view that astrocytes are together with neurons important components of functional neural networks. Dr. Barca-Mayo and colleagues examined the role that astrocytes play in the generation of circadian rhythms by the suprachiasmatic nucleus. They used a conditional *Glast-Cre, Bmal1^{flx/flx}* expressing mice to selectively knock down *Bmal1* in a subset of astrocytes and examine how disruption of this circadian clock gene in astrocytes would affect behavioral and gene expression circadian rhythms. The results provide strong evidence for a role of *Bmal1* expression in cortical and hippocampal neurons altering the performance on behavioral tasks. A major limitation of the manuscript is that the conclusions and descriptions of the behavioral data do not match the data shown.*

Authors' Answer. We thank the reviewer for appreciating our study and for the critical and constructive feedbacks. In particular, we are grateful that the reviewer acknowledges that "The results provide strong evidence for a role of *Bmal1* expression in ...altering the performance on behavioral tasks". **We estimate having addressed all the raised comments and we hope that the new version of the MS satisfies the reviewer.** The revised text is highlighted in red in the MS. The point-by-point response follows.

Reviewer's Major comment 1. *In Figure 1a there are three populations, GFAP-expressing astrocytes, Tomato-expressing astrocytes and those that express both GFAP and Tomato. Also, BMAL1 was expressed in only 43.7% of GFAP-expressing cells and reduced in ~60% of this subset of cells so that ~18% of astrocytes continue to express *Bmal1*. These data raise a concern about whether the identity of the specific cells with knocked-out *Bmal1* can be determined. Why is *Bmal1* not reduced in all *Glast-Cre* cells? Does *Glast-CreERT2* drive recombination in S100-beta-positive or glutamate synthase-positive astrocytes in the SCN.*

Authors' Answer. To address your concern, and also by following suggestions of reviewer #1 (See answer to reviewer 1 comment 1), we have now quantified the co-localization of Td-Tomato positive cells with the astrocyte markers GFAP and S100 β and also quantified *Bmal1* depletion in Td-Tomato positive and Td-Tomato negative cells (new Fig. 1 and supplementary Fig. 1 of the revised MS).

The deleter mouse line used in our study (knock-in of CRE-ERT2 gene into the endogenous *Glast* locus) has been previously characterized by Magdalena Götz (Mori T, Tanaka K, Buffo A, Wurst W, Kühn R, Götz M, Inducible gene deletion in astroglia and radial glia- a valuable tool for functional and lineage analysis. *Glia*. 2006. 54, 21-34). The percentage of co-localization among GFAP or S100 β and *Glast* positive cells that we detected in the SCN of our mice is not very different from the one reported by Magdalena Götz in cortex or striatum of adult mice (~80 and ~70%, respectively).

We found that in the SCN of our mutant mice, depletion of *Bmal1* occurs in the majority of the astrocytes (i.e. 60% of Td-Tomato positive cells that co-localized with GFAP or S100 β). Despite we follow the same tamoxifen treatment described by Magdalena Gotz to optimize Cre activity induction in the adult brain, there might regional differences in the recombination efficiency among different brain areas. This might explain the lack of recombination in some of the *Glast* positive cells in the SCN. However, our results indicate that deletion of *Bmal1* in the majority of *Glast* positive astrocytes is sufficient to drive a global down-regulation of BMAL1 protein in the SCN (see new Fig.1 and supplementary Fig.1), consistent with our hypothesis that deletion of *Bmal1* in astrocytes might affect the neuronal clock.

Reviewer's Major comment 2. *The description of the running wheel activity does not match the data shown in Figure 2. The authors indicate that there is a 1.31 hour phase advance of the wheel running activity in the *Bmal1*cKO mice. There is no phase advance apparent in the data shown in Fig. 2b. The*

blue lines in the average activity graphs show the onset of activity in control and Bmal1cKO mice to be very similar or possibly the activity in the Bmal1cKO mice start a little later, which would indicate a altered phase angle of entrainment not a phase change.

Authors' Answer. Thanks for raising this important point that otherwise could have been source of misunderstandings. We have now modified the text accordingly, modified the representation of the total average activity (new Fig. 2c) and added: **i)** the periodogram (to show the bimodal pattern of activity (new Fig 2b, right panels); **ii)** the activity onset in DD showing that Bmal1cKO mice delayed their active phase as compared to control animals (new Fig 2d, left panel); **iii)** onset and offset of activity in rLD showing that Bmal1cKO mice advanced the onset and offset in the wheel running activity as compared to control animals (new Fig 2d, middle and right panels).

We found no significant differences between Bmal1cKO and control animals in the onset of activity in LD cycles (blue lines) (7.08h vs 7,61h for Bmal1cKO mice, paired t-test, p=0.54)

Reviewer's Major comment 3. *The rate of re-entrainment is not directly studied and the data presented provide no support for the claim that the rate of re-entrainment is affected.*

Authors' Answer. Thanks for pointing this out. We agree that we haven't performed advances or delays of light-dark cycles to determine directly the rate of re-entrainment in Bmal1cKO animals. We have used the word "re-entrainment" to describe that Bmal1cKO mice were not able to suppress their activity under the new light-dark cycle and thus were not able to "entrain again" or to synchronize their internal clock to the light. We acknowledge the concern of the referee and have now changed the term "re-entrainment" in "adjust the activity to the new light-dark cycle".

Reviewer's Major comment 4. *Given the small effects of the Bmal1 knock-out on the locomotor activity the in vivo effects of the picrotoxin or PTZ injections are hard to interpret because they may just reflect the variability of the running wheel behavior of individual mice.*

Authors' Answer. We agree with the referee that the circadian locomotor behaviour of Bmal1cKO mice is mild. We specify in the "Results" section (page 5) that "Any of Bmal1cKO mice showed loss of rhythms, suggesting that Bmal1 ablation in SCN astrocytes has a mild impact on the clock".

To address the referee concern we have increased the number of animals that were treated with PTZ and subjected to the circadian locomotor test (control n=9 and Bmal1cKO n=10). These new experiments allowed us to increase the statistical significance of our experiment and minimize potential individual variabilities. As shown in the new Fig.7 of the revised MS, we found that PTZ treatment rescues the locomotor behaviour in Bmal1cKO mice.

Reviewer's Major comment 5. *No experiments were performed nor data shown that is consistent with the statement "suggesting a defect in their masking responses".*

Authors' Answer. Thanks for raising this important point that otherwise could have been source of misunderstandings. We used the term "masking" to describe that the activity of Bmal1cKO mice was not suppressed by the light after release from constant darkness to a new LD cycle (rLD). We agree with the referee that we have not performed the experiments to determine the masking response of our mutants and we have now removed this phrase in the revised MS.

Reviewer's Major comment 6. *Bmal1 is a canonical clock gene and an important transcription factor. One of the challenges in studying Bmal1 knock-outs is to distinguish which observed effects are due to circadian clock disruption and which are due to disruption of gene transcription. The effects on learning and memory may result in a disruption of astrocyte signalling in the cortex and hippocampus that is independent of any disruption of a circadian clock in the astrocytes.*

Authors' Answer. We certainly agree with the logic of the reviewer. Indeed, we also stated in the "Introduction" section that "oscillations in abundance of those core circadian clock proteins in brain and peripheral tissues drive a cascade of transcriptional rhythms of output genes that are not involved in the timekeeping mechanism itself, but underlie local behavioural and physiological process". In the "Discussion" we specify that "...a major challenge, which holds true for most of the studies involving

genetic manipulations of core clock genes, involves distinguishing the specific importance of circadian oscillation versus the “static” function of clock genes”...

On the other hand, Colas et al., found that the GABAA receptor antagonist must be administered at specific times of the day to revert the cognitive phenotype in their model. We followed the protocol they described (treatment with PTZ or PTX at ZT6) and found a rescue on the memory phenotypes in our mice. This might suggest a circadian-related effect of Bmal1 rather than static. The importance of rhythmic versus static function of Bmal1 is, definitely, a relevant scientific challenge in the field, but it is beyond the scope of our study. As we specify in the “Discussion” section, *“Despite this fact, our results together with the elegant studies in Drosophila^{5,6}, not only emphasize the conservation of cellular and molecular mechanisms that regulate behavior in mammals and insects but also demonstrate the capacity for astrocyte-to-neuron signaling in the circadian circuitry”*.

D Colas, B Chuluun, D Warriar, M Blank, D Z Wetmore, P Buckmaster, C C Garner, and H C Heller. Short-term treatment with the GABAA receptor antagonist pentylentetrazole produces a sustained pro-cognitive benefit in a mouse model of Down's syndrome. *Br J Pharmacol.* 2013. 169, 963–973.

Reviewer's Major comment 7. *The behavioral data should be analyzed with a periodogram to clearly demonstrate the presence of the bimodal behavior.*

Authors' Answer. As suggested, the periodograms are shown in the new Fig. 2 of the revised MS.

Reviewer's minor concerns

-1. *The meaning of the following sentence is unclear. “During the light-dark condition, Bmal1cKO mice showed similar robust circadian locomotor activity than control mice (Fig. 2b), with no differences in the periodicity, neither in the total average activity (Fig. 2c).”*

Authors' Answer. We have revised wording in the MS, in particular in the result section of the revised MS we now write: *“During the light-dark (LD) condition, the locomotor activity of Bmal1cKO mice was indistinguishable from that of control animals (Fig. 2b left panels, Fig. 2c), showing no differences in the periodicity or in the total average activity (supplementary Fig. 2)”*.

-2. *“In contrast, VIP expression was not downregulated in the SCN of Bmal1cKO mice at ZT12 (Fig. 3b) suggesting constitutive repression of VIP expression by astrocytic Bmal1 leading to impairment of circadian rhythmicity of VIP in the mutant mice.” I suggest taking constitutive out of this sentence since the Bmal1 expression is rhythmic and the data suggest that the Bmal1 rhythm contributes to the VIP rhythm.*

Authors' Answer. We have now removed “constitutive” from the sentence.

-3. *On pg. 3 - reference 28 is not the correct reference.*

Authors' Answer. We have revised the bibliography and added the correct reference for the description of Bmal1 floxed mouse

-4. *Reference 18 demonstrates a strong impairment on the novel recognition test in Bmal1 knockout mice.*

Authors' Answer. The study (Wardlaw et al., 2014, Reference 19 of the revised MS) reported that Bmal1 knockout has no deficits in novel object recognition (and even enhancement 24 h later compared to controls). However, in this study the authors found that Bmal1 knockout are impaired in hippocampus-dependent memory task (fear conditioning and Morris water maze) as we specify in the “Results” section (Page 5), *“Perturbations of circadian rhythms in humans, such as in shift workers and resulting from jet lag¹⁸, or in constitutive knockout mice for clock genes such as Bmal1^{-/-}, have been associated with cognitive dysfunction¹⁹”*. We discuss that Bmal1cKO displayed differences with Bmal1 knockout mice that can be due to the use of acute mouse model, since we avoid structural, or developmental functional abnormalities or compensation. We realized that this sentence could have been a source of misunderstandings. Thus in the revised MS we have removed this comparison from the second paragraph of the discussion.

We are grateful for the constructive criticism and helpful suggestions. We sincerely hope you will agree that the revised manuscript is ready for publication in Nature Communications now.

Reviewers' comments:

Reviewer #1 (Remarks to the Author):

My congratulations to the authors. My initial concerns have been addressed.

Reviewer #2 (Remarks to the Author):

The authors have significantly improved the manuscript and addressed my questions.

Reviewer #3 (Remarks to the Author):

The current manuscript describes an impressive set of studies that examined the role of Bmal1 expression in a subset of astrocytes and the role these astrocytes play in regulating circadian and cognitive behavior. The authors were very responsive to the previous reviews and have added a significant amount of new data demonstrating the role of astrocytes in regulating circadian activity by regulating activity at GABAergic synapses. The revisions have significantly improved the manuscript although a few issues remain.

Major points:

1. The description of the wheel running activity analysis still remains unclear. In line 136 the authors note that the locomotor activity of the Bmal1cKO mice was not inhibited in the light phase. In the actograms shown in Figure 2 and supplemental Figure 7, there is no evidence for increased activity in the light phase. However, the activity waveforms shown in Figure 2 c show a significant amount of activity in the light phase.
2. The VIP rhythm is driven by the light-dark cycle not the circadian clock. This suggests the mechanism of disruption of the VIP rhythm is different than the reduction in the clock gene rhythms.

Minor points:

1. In line 106 do the authors mean that the magnitude of the BMAL expression was reduced 70% or the number of Td-Tomato cells expressing BMAL1 was reduced by 70%.
2. Line 115 "ought" should be "sought"
3. Line 56 "over" should be deleted.
4. In some places the decimal sign is a period and in others it is a comma.
5. Lines 128 & 129 the SEM values are given with 5 significant digits. The bin size of the wheel running activity is 5 minutes which produces a resolution less than five digits.
6. Line 365 - " any" should be replaced with "none"
7. Line 371 - "nowadays" should be replaced with "todays"

Reviewer #1 (Remarks to the Author):

My congratulations to the authors. My initial concerns have been addressed.

Authors' Answer. We thank the reviewer for his/her strong support of this study.

Reviewer #2 (Remarks to the Author):

The authors have significantly improved the manuscript and addressed my questions.

Authors' Answer. We thank the reviewer for his/her strong support of this study

Reviewer #3 (Remarks to the Author):

The current manuscript describes an impressive set of studies that examined the role of Bmal1 expression in a subset of astrocytes and the role these astrocytes play in regulating circadian and cognitive behavior. The authors were very responsive to the previous reviews and have added a significant amount of new data demonstrating the role of astrocytes in regulating circadian activity by regulating activity at GABAergic synapses. The revisions have significantly improved the manuscript although a few issues remain.

Authors' Answer. We thank the reviewer for the critical and constructive feedbacks, which helped us to improve the quality of our study. In particular, we are grateful that the reviewer acknowledges that *"The current manuscript describes an impressive set of studies.."* and that *"The authors were very responsive to the previous reviews and have added a significant amount of new data demonstrating the role ..."*. We estimate having addressed your concerns and we hope that the new version of the MS now satisfies you. The revised text is highlighted in red in the MS. The point-by-point response follows.

Reviewer's Major comment 1. *The description of the wheel running activity analysis still remains unclear. In line 136 the authors note that the locomotor activity of the Bmal1cKO mice was not inhibited in the light phase. In the actograms shown in Figure 2 and supplemental Figure 7, there is no evidence for increased activity in the light phase. However, the activity waveforms shown in Figure 2 c show a significant amount of activity in the light phase.*

Authors' Answer.

We thank the referee for pointing this out. We agree that the phrase on line 136 "locomotor activity was not inhibited during the light period" (referring to Bmal1cKO mice) might lead to misunderstandings, since Bmal1cKO mice just display advanced onset of activity, but not increased activity during the whole light period. Thus, we removed this phrase from the revised MS to avoid any confusion.

In the previous version of the MS, we represented the activity distribution in rLD cycle as the average amount of activity in 5-minute bins over each animal's circadian period. The same analysis was performed for the Fig7C (rLD cycle), although we did not specify it in the figure legend. The analysis was performed in such a way because of the significant differences in the onset of activity of the mutants, as well as the trend of decreased period of Bmal1cKO as compared to control animals, in this lighting condition ($23,88 \pm 0,1058$ vs $24,15 \pm 0,1787$, $p=0.24$). According to your suggestion, we have now reanalyzed the activity waveforms for the rLD in Fig2C and Fig7C. These data are now plotted with night-time hours, from 7 to 19 and given in Zeitgeber time (ZT), such that ZT0 (lights-on) = hour 19.

As we comment in the previous point by point to the referees (specifically, answer to Referee 1, comment 11), PTZ is known to shift the phase of activity, although the mechanism is currently unknown. Indeed, we observed a similar trend in both control and Bmal1cKO mice (see Fig 7C). Remarkably, no differences were found among genotypes suggesting a direct effect of the drug and not of the genotype. On the other hand, the protocol treatment (at ZT6) that we follow, previously described by Ruby and Colas, might be the cause of this shift due to the arousal effect of the injections, which is known to stimulate activity when injections are given repeatedly.

Reviewer's Major comment 2. *The VIP rhythm is driven by the light-dark cycle not the circadian clock. This suggests the mechanism of disruption of the VIP rhythm is different than the reduction in the clock gene rhythms.*

Authors' Answer.

We have taken into account the comment of the reviewer and we have now rephrased the text that was suggesting that the impaired VIP oscillation in Bmal1cKO animals was due to the impaired circadian clock (line 158, 162-169, 173, 185). Indeed, we have now specifically added this concern on the mechanism by which astrocytic BMAL alters the VIP rhythm in the discussion of the revised version of the MS (line 418).

Reviewer's minor concerns

1. *In line 106 do the authors mean that the magnitude of the BMAL expression was reduced 70% or the number of Td-Tomato cells expressing BMAL1 was reduced by 70%.*

Authors' Answer. Thank you for pointing this out. We have clarified that the number of Td-Tomato cells expressing BMAL1 was reduced by 70%.

2. *Line 115 "ought" should be "sought"*

Authors' Answer. We have changed "ought" by "sought"

3. *Line 56 "over" should be deleted.*

Authors' Answer. We have deleted "over"

4. *In some places the decimal sign is a period and in others it is a comma.*

Authors' Answer. Thank you for pointing this out. We have reviewed the manuscript and we have uniformed by using the decimal sign period.

5. *Lines 128 & 129 the SEM values are given with 5 significant digits. The bin size of the wheel running activity is 5 minutes which produces a resolution less than five digits.*

Authors' Answer. The periodicity in lines 128&129 is reported in hours and not in minutes and thus the conversion might lead to 5 significant digits (for example if the running period of a mice is 1435 minutes (287, 5-min bin), when converted to hours it is 23.91667). We have now uniformed the number of decimal digits in the MS and the values are given with 2 decimal digits for mean values and EEM values.

6. *Line 365 – "any" should be replaced with "none"*

Authors' Answer. We have changed "any" by "none".

7. *Line 371 – "nowadays" should be replaced with "today's"*

Authors' Answer. We have changed "*nowadays*" by "*today's*".

REVIEWERS' COMMENTS:

Reviewer #3 (Remarks to the Author):

The authors have answered all of my concerns and I have no additional issues.